

# Air – snow exchange of nitrate: a modelling approach to investigate physicochemical processes in surface snow at Dome C, Antarctica

Josué Bock[1], Joël Savarino[2,3], and Ghislain Picard[2,3]

[1]Centre for Ocean and Atmospheric Sciences, School of Environmental Sciences, University of East Anglia, Norwich Research Park, Norfolk, NR4 7TJ, Norwich, UK
[2]Université Grenoble Alpes, Laboratoire de Glaciologie et Géophysique de l'Environnement (LGGE), 38041 Grenoble, France
[3]CNRS, LGGE UMR5183, 38041 Grenoble, France

*Correspondence to:* Josué Bock (j.bock@uea.ac.uk)

**Abstract.** Snowpack is a multiphase (photo)chemical reactor that strongly influences the air composition in polar and snow-covered regions. Snowpack plays a special role in the nitrogen cycle, as it has been shown that nitrate undergoes numerous recycling stages (including photolysis) in the snow before being permanently buried in the ice. However, the current understanding of these

physicochemical processes remains very poor. Several modelling studies have attempted to reproduce (photo)chemical reactions inside snow grains, but these required strong assumptions to characterise snow reactive properties, which are not well defined. Air – snow exchange processes such as adsorption, solid state diffusion or co-condensation also affect snow chemical composition. Here, we develop a model including a physically based parameterisation of these processes for nitrate. Us-

ing as input a one-year long time series of atmospheric nitrate concentration measured at Dome C, Antarctica, our process-resolving model reproduces with good agreement the nitrate concentration measured in surface snow. By investigating the relative importance of the main exchange processes, this study shows that, on the one hand, the combination of bulk diffusion and co-condensation incorporation processes allows a good reproduction of the measurements (correlation coefficient

$r = 0.95$), with a correct amplitude and timing of summer peak concentration of nitrate in snow. During wintertime, nitrate concentration in surface snow is mainly driven by thermodynamic equilibrium, whilst the peak observed in summer is explained by the kinetic process of co-condensation. On the other hand, the adsorption of nitric acid on the surface of the snow grains, constrained by an already existing parameterisation for the isotherm, fails to fit the observed variations. During win-

ter and spring, the modelled adsorbed concentration of nitrate is 2.5 and 8.3-fold higher than the measured one, respectively. A strong diurnal variation driven by the temperature cycle and a peak occurring in early spring are two other major features that do not match the measurements. This study clearly demonstrates that the co-condensation process is the most important to explain nitrate incorporation in snow subject to temperature gradients. The parameterisation developed for this pro-



cess can now be used as a foundation piece in snowpack models to predict the inter-relationship between snow physical evolution and snow nitrate chemistry.

## 1   Introduction

The nitrogen cycle governs atmospheric oxidants budget through the photochemistry of nitrogen oxides ($NO_x$=$NO$ + $NO_2$) which are strongly coupled with ozone ($O_3$) and hydroxyl ($OH$) chem-

istry in the troposphere (Seinfeld and Pandis, 1998; Finlayson-Pitts and Pitts, 2000). Atmospheric nitrate is the end product of $NO_x$ oxidation, and the snowpack (and subsequently the firn and ice) act as a sink for it. Temporal variations of the nitrate concentration recorded in ice cores (Legrand and Mayewski, 1997) could thus provide information about the oxidative capacity of the atmosphere in past times (Dibb et al., 1998), or even about past solar activity (Traversi et al., 2012). However,

as illustrated by Davis et al. (2008, Fig. 2), several post-deposition processes occur in the snow and hamper our current ability to interpret ice core records of nitrate. As a first evidence of such post-deposition processes, $NO_x$ has been shown to be produced in sunlit snowpack (Honrath et al., 1999, 2000b, 2002; Jones et al., 2000; Beine et al., 2002), with a production pathway involving nitrate photolysis in snow rapidly elucidated afterwards (Jones et al., 2000; Dibb et al., 2002; Hon-

rath et al., 2002). These pioneering works drove numerous field campaigns (e.g. SNOW99 (Honrath et al., 2000b), ISCAT2000 (Davis et al., 2004), ANTCI (Eisele et al., 2008), CHABLIS (Jones et al., 2008), OPALE (Preunkert et al., 2012)), as well as laboratory studies (Honrath et al., 2000a; Dubowski et al., 2001, 2002; Chu and Anastasio, 2003, 2007; Cotter et al., 2003; Zhu et al., 2010; Meusinger et al., 2014; Berhanu et al., 2014) and modelling studies (Jacobi and Hilker, 2007; Boxe

and Saiz-Lopez, 2008; Liao and Tan, 2008; Bock and Jacobi, 2010; Thomas et al., 2011; Toyota et al., 2014; Erbland et al., 2015; Murray et al., 2015) in order to improve the understanding of the underlying processes responsible for the nitrogen recycling inside the snowpack. These studies focused on the nitrate photolysis in the photic zone of the snowpack and the subsequent release of $NO_x$ to the overlying atmosphere. However, none of these studies investigated the physicochemical incor-

poration processes of atmospheric nitrate into snow. Yet, it is now well documented that the nitrate concentration in the surface snow features a seasonal peak during summer on the Antarctic plateau (Erbland et al., 2013, and ref. therein), when the solar flux is close to its annual maximum and photolysis is strongest. This shows that some incorporation processes counteract photochemical loss, and thus need to be studied in order to understand the nitrate budget of the snow. In a recent study

from Jones and co-workers, measurements of gaseous $HNO_3$ were carried out with a high temporal resolution of 10 min, during 4 winter months at Halley station, located at coastal Antarctica (Jones et al., 2014). This work reveals that $HNO_3$ concentration is highly correlated with the temperature, highlighting that physical air–snow exchange processes play a key role during this period of the year.



Numerous experimental studies of adsorption on ice surfaces have demonstrated that several
chemical compounds, and especially acidic gases such as HCl and HNO$_3$, have a great affinity
for ice surface (see reviews by Abbatt, 2003; Huthwelker et al., 2006). Other post-deposition physi-
cal processes also affect snow chemical composition (Dominé et al., 2008). Several small chemical
species, such as HCl (Dominé et al., 1994; Thibert and Dominé, 1997), HNO$_3$ (Thibert and Dom-
iné, 1998), HCHO (Perrier et al., 2003; Barret et al., 2011b) and H$_2$O$_2$ (Sigg et al. (1992) and ref.
therein; Conklin et al. (1993); Jacob and Klockow (1993); McConnell et al. (1997b)) form solid
solutions in ice. Other species such as SO$_2$ (Clapsaddle and Lamb, 1989; Conklin and Bales, 1993;
Huthwelker et al., 2001) and HONO (Pinzer et al., 2010; Kerbrat et al., 2010) may also diffuse into
bulk ice. Thus, solid state diffusion is able to bury these molecules in the inner parts of snow crystals,
or on the contrary to make it available for surface (photo)chemical reactions after migration from
the bulk crystal to its surface.

Another physical process, known as co-condensation, consists in the simultaneous condensation
of water vapour and trace gases at the air–ice interface. Water vapour fluxes in the snowpack are
mainly driven by temperature gradients, leading to massive mass transfer from the warmest snow
layers which sublimate, towards the coldest parts where vapour condensates (Calonne et al., 2014;
Ebner et al., 2015; Hansen and Foslien, 2015). More generally, the subsequent change in snow
morphology, called temperature gradient metamorphism, affects the whole snowpack following sea-
sonal temperature variations (Marbouty, 1980; Sommerfeld, 1983; Flin and Brzoska, 2008; Pinzer
and Schneebeli, 2009; Pinzer et al., 2012; Ebner et al., 2015), and particularly the upper part of
the snowpack subjected to the diurnal temperature cycles (Picard et al., 2012; Champollion et al.,
2013, and ref. therein). Indeed, high crystal growth rates are observed at the surface of the snow-
pack, and at about 10 cm under the snow surface (Colbeck, 1989, Fig. 8) though the exact depth is
subject to debate (Kuipers Munneke et al., 2009; Libois et al., 2014). Along with the vapour flux,
trace impurities present in the interstitial air, or temporarily adsorbed on the ice surface, might be
incorporated inside the crystals (Conklin et al., 1993; Bales et al., 1995; Dominé and Thibert, 1996;
Xueref and Dominé, 2003; Dominé and Rauzy, 2004; Kärcher and Basko, 2004; Ullerstam and Ab-
batt, 2005; Kärcher et al., 2009). This kinetic process of incorporation is much more efficient than
air–ice thermodynamic equilibrium, which probably explains why measured concentrations have
sometimes been shown to be of out of equilibrium (Bales et al., 1995; Dominé and Thibert, 1995,
1996; Ullerstam and Abbatt, 2005).

The models of snow chemistry developed so far focus on snow-to-air processes driven by (photo)chemistry,
since they mainly intended to reproduce field measurements of NO$_x$ fluxes emitted by the snowpack.
Recent work from Erbland et al. (2013, 2015) indeed suggest that the denitrification of the snowpack
by means of physical release is negligible compared to the photochemical processes. Thus, air-to-
snow physical exchange processes were ignored in several studies (Boxe and Saiz-Lopez, 2008;
Bock and Jacobi, 2010). In other models, these processes were bypassed through ad-hoc param-



eterisation and/or implemented using air–liquid equilibrium following Henry's law, based on the assumption that snow crystals are covered by a liquid layer (Liao and Tan, 2008; Thomas et al., 2011; Toyota et al., 2014). Such modelling approaches and their pitfalls were discussed in details by Dominé et al. (2013). As far as we are aware, the only physically based modelling studies of air–snow exchange processes were carried out at the late 1990's to interpret multiyear firn concentration profiles of $H_2O_2$ (McConnell et al., 1997a, b, 1998) and of HCHO (Hutterli et al., 1999, 2002). As summarised by Hutterli et al. (2003, Fig. 1), these two series of modelling studies handled air-snow uptake/release through an exchange coefficient accounting for an Henry's law type partitioning between the two compartments, but did not included the co-condensation process nor the solid state diffusion inside the ice crystals. More recently, Barret et al. (2011a) proposed an air–snow exchange model to reproduce surface snow HCHO concentration. In this study, the surface snow is depicted as a unique spherical, layered grain whose surface concentration of HCHO is constrained by the air–ice thermodynamic equilibrium. Their model uses as input the measured gas phase HCHO concentration and solves the spherical diffusion equation with radial symmetry to calculate the mean concentration in the whole snow grain. Their results reproduce the concentration measured in surface snow during a 36-hour intensive sampling period in the course of OASIS 2009 campaign with fairly good agreement (Barret et al., 2011a, Fig. 4).

For the first time, we propose a process-resolving model for air–snow exchange of nitric acid ($HNO_3$), which allows an investigation of the above mentioned physicochemical exchange processes. Following a similar approach to that of Barret et al. (2011a), we developed a model considering a single spherical layered snow grain. This snow grain is assumed to be in direct contact with the air just above the snowpack. Using the atmospheric nitrate concentration measured at Dome C (DC) for about one year as input, the model calculates the snow nitrate concentration resulting from (i) adsorption on the snow grain surface, (ii) solubilisation into the outermost layer according to thermodynamic equilibrium, and diffusion inside the snow grain and (iii) co-condensation following vapour fluxes inside the upper snowpack. Model results are compared to nitrate concentration in the uppermost $\sim 4$ mm of the snowpack ("skin layer" hereafter).

The input datasets are presented in the next section, and the model is described in Sect. 3. The results obtained in configuration 1 (adsorption only) are presented and discussed in Sect. 4, and those relative to the model configuration 2 (solid state diffusion) are presented in Sect. 5.





## 2 Input data description

### 2.1 Annual atmospheric and skin layer nitrate concentrations at Dome C

#### 2.1.1 Atmospheric nitrate

Atmospheric nitrate, which includes both particulate nitrate and gaseous $HNO_3$, was measured con-
tinuously at DC between January 2009 and January 2010 using a high-volume air sampler placed
5 m above the snow surface (Erbland et al., 2013). Atmospheric nitrate was collected on glass fibre
filters, which efficiently trap both particulate nitrate and gaseous $HNO_3$ (Frey et al., 2009; Erbland
et al., 2013). Atmospheric nitrate was quantitatively extracted in $40\,cm^3$ of ultrapure water via cen-
trifugation using Millipore Centricon™ filter units, and its concentration was then determined using
the colorimetric method as described in Erbland et al. (2013). Atmospheric nitrate concentration was
calculated as the ratio of the total $NO_3^-$ filter loading to the total volume of air pumped through the
filter at STP conditions and expressed in $ng\,m^{-3}$.

Atmospheric nitrate samples were collected for 37 separate 5–7 day periods (see Fig. 1a). Over the
year, 10 samples were dedicated to $^{35}S$ measurement. The missing values were linearly interpolated
hereafter (dashed lines in Fig. 1a). As can be seen in Fig. 1a, atmospheric nitrate concentration is low
and steady, with a mean value of $(8.2 \pm 5.1)\,ng\,m^{-3}$ from March to September, followed by a sharp
increase during the spring (average value of $(98.5 \pm 39.7)\,ng\,m^{-3}$ from October to December, with
peak values greater than $130\,ng\,m^{-3}$). A rapid decrease is observed in early summer. This yearly
pattern is in good agreement with previous measurements performed at DC between January 2007
and January 2008 (Frey et al., 2009).

A few simultaneous measurements of atmospheric nitrate (also reported as "filterable nitrate",
$f-NO_3^-$) and $HNO_3$ allow to gain further insight into the partitioning between both. Arimoto et al.
(2008, Fig. 5) and Davis et al. (2008, Fig. 3) report concurrent measurements of $f-NO_3^-$ and $HNO_3$
carried out during 23 days in the course of the ANTCI campaign, at South Pole. Atmospheric nitrate
was measured in a very similar way as at DC, using a high-volume air sampler with Whatman 41™
filters which have been shown to efficiently collect atmospheric nitrate as well (Arimoto et al., 2008,
and ref. therein). This dataset reveals that $HNO_3$ accounts for the major part of the atmospheric
nitrate over the whole period of measurements, and we calculated an average proportion of 80 % of
$HNO_3$ among total $f-NO_3^-$ (Davis et al., 2008, Fig. 3).

Over the 2009–2010 period, $HNO_3$ was measured at DC using annular denuder tube, with 48
sampling periods of 2.5 days on average (unpublished, personal communication, B. Jourdain and M.
Legrand, 2012). These different sampling periods between the data sets hinder our ability to make
a close comparison, but it is obvious that both times series show a very good agreement (data not
shown). The ratio of $HNO_3$ to atmospheric nitrate is of the same order as that obtained at South
Pole.





Another recent study presented a multi-year record of particulate nitrate at DC, collected on low volume sampler with Teflon filters (Traversi et al., 2014). Both the absolute nitrate concentration and the overall temporal pattern reported in that study are in good agreement with those of Erbland et al. (2013). By comparing the measurements of an 8-stage impactor along with those provided by a PM10 device, the authors conclude that during late summer (January and February), only 12.5 % of atmospheric nitrate is collected on PM10 PTFE filters, while this fraction reach 30 % for the November and December months. Thus, further accounting for a temporal variation in the partitioning between gaseous $HNO_3$ and particulate nitrate is needed to accurately retrieve $HNO_3$ concentration from atmospheric nitrate measurement.

To conclude, atmospheric nitrate measured at DC during several years using different methods features a very consistent and reproducible temporal pattern. Further comparisons between gaseous and particulate fractions indicate that $HNO_3$ accounts for the major part of atmospheric nitrate. For sake of simplicity, we assume hereafter that the concentration of gaseous $HNO_3$ used as input in our model is equal to the concentration of atmospheric nitrate. This assumption will be further discussed along with the results of the model.

### 2.1.2 Snow nitrate

Nitrate concentration has been measured year round between 2008 and 2010 during NITE DC program (NITrate Evolution in surface snow at Dome C). The skin layer (estimated average thickness of $(4 \pm 2)$ mm) was sampled once or twice a day during summer, and about once a week during winter (Erbland et al., 2013). The uncertainty ascribed to spatial variability and sampling method is estimated to be 20 %. In this study, we only used data from 30 January 2009 to 31 January 2010. This data set was already published (Erbland et al., 2013, Fig. 6), and is reproduced in Fig. 1a. $NO_3^-$ concentration in the skin layer exhibits a seasonal pattern similar to that of atmospheric nitrate: it remains relatively low and steady during winter, with an average value of $(161 \pm 50)$ ng g$^{-1}$ during the polar night, i.e. from March to September. Then, a sharp increase occurs around mid-November, with concentration in the 600–1400 ng g$^{-1}$ range. The temporal lag of 3–4 weeks between the atmospheric and skin layer variations indicates a complex air–snow transfer function, that this work aims at elucidating using a process-resolving model.

Further measurements of snow nitrate concentration were carried out in snow pits at DC, up to 50 cm and every 6 weeks on average in winter (Fig. 2a), and up to 20 cm depth and every week in summer (Fig. 2b). These concentration profiles of the upper snowpack give a better insight of the nitrogen recycling occurring in the snow. Indeed, the highest concentration measured in summer is located in the top few mm to cm, whilst nitrate concentration dramatically decreases at greater depths, never exceeding 110 ng g$^{-1}$ below 5 cm depth. During winter, the top 15–20 cm of the snowpack replenish in nitrate, with concentration in the 200–400 ng g$^{-1}$ range, whilst the deepest layer concentration decreases to less than 50 ng g$^{-1}$. This advocates for a huge seasonal remobil-





isation of nitrate in the snowpack, with upwards transfer during spring and summer and slower, smoother downwards transfer during winter.

These temporal variations of $NO_3^-$ observed in DC surface snow are also similar to the general
trends featured by previous measurements in surface snow made at Halley station in coastal Antarctica from March 2004 to February 2005 (Wolff et al., 2008; Jones et al., 2011).

### 2.2  Snowpack physical properties

#### 2.2.1  Snow temperature

Snow temperature is a key parameter for modelling snow chemistry since all processes implied in
snow chemical exchange are temperature dependent. In addition, snow metamorphism and water vapour flux depend on temperature as well as on the vertical gradient of the temperature profile (see for instance Marbouty, 1980; Sommerfeld, 1983; Colbeck, 1989; Flin and Brzoska, 2008). We used modelled data to get snow surface temperature over the whole year of nitrate measurements.

A snowpack thermal diffusion model including a surface scheme coupled with a radiative transfer
model to account precisely for the absorption of the radiation inside the snowpack is used. The snowpack is discretised in horizontally homogeneous layers whose thickness increase exponentially with depth. The model takes as input meteorological forcing from ERA-Interim reanalysis and computes the evolution of temperature profile (Picard et al., 2009). Predictions were successfully compared to daily passive microwave satellite data, and a comparison with Brun et al. (2011) results shows good
skills.

We used the modelled temperature in the uppermost 3 mm thick layer (which is also the surface "skin" temperature used in the surface energy budget calculation) and apply linear interpolation to down-scale the hourly data to 10 min, the timestep of our model. The modelled snow surface temperature is shown in Fig. 1b.

We compared the modelled temperature with skin temperature deduced from BSRN (baseline surface radiation network) upwelling longwave radiation observations (Christian Lanconelli, personal communication; see SI 2). From this 3 month data set (from November 2009 to January 2010, raw data), the comparison revealed a small warm bias of the model ($\sim 2.5$ K), and a slight underestimation of the amplitude of the diurnal cycle (see SI 2) which agrees with other studies using
ERA-Interim (Fréville et al., 2014). However, since this comparison was only possible during the summertime, the same discrepancies between modelled and measured temperatures would not necessarily hold in winter.

#### 2.2.2  Specific surface area

In our model, the physical description of the snow mainly relies on the snow specific surface area
(SSA) value, which directly affects exchanges through the air–snow interface (see for example Dom-





iné et al., 2008). Assuming spherical grains, the radius follows the relation:

$$R = \frac{3}{SSA \times \rho_{\mathrm{ice}}}$$  (1)

where $R$ is the radius (in m), $SSA$ is the snow specific surface area (in $\mathrm{m^2\,kg^{-1}}$) and $\rho_{\mathrm{ice}}$ is the ice density, with $\rho_{\mathrm{ice}} \simeq 924\,\mathrm{kg\,m^{-3}}$ (Hobbs, 1974, at -50 °C, DC annual mean temperature). When this study was initiated, the only SSA value reported at DC was $38.1\,\mathrm{m^2\,kg^{-1}}$ for the first centimetre, decreasing monotonically to $13.6\,\mathrm{m^2\,kg^{-1}}$ at 70 cm depth (Gallet et al., 2011, Fig. 4 and Table A1). Recent work specifically studying surface hoar at DC reported very close values, with an average of $39.0\,\mathrm{m^2\,kg^{-1}}$ for the top first centimetre of snow, and $26.4\,\mathrm{m^2\,kg^{-1}}$ for the second centimetre (Gallet et al., 2014). Thus, SSA was set to a value of $38.1\,\mathrm{m^2\,kg^{-1}}$ by default in the model, leading to a grain radius $R = 85\,\mu\mathrm{m}$. Recently Libois et al. (2015) investigated seasonal variations of SSA at DC showing that these values are typical of the summer while 2 to 3-fold higher values are observed in winter. The effect of changing SSA was further tested in a sensitivity test presented in Sect. 5.4.

## 3   Model description

### 3.1   From gaseous $HNO_3$ to solid solution of nitrate in snow

A brief summary of the current knowledge about solvation steps which lead gaseous $HNO_3$ to form solid solution in bulk ice is presented in this section.

The uptake of trace gases on ice, and more specifically of acidic gases among which $HNO_3$, has been the subject of considerable investigation (see reviews by Abbatt, 2003; Huthwelker et al., 2006). Conceptually, this uptake proceeds firstly by molecular adsorption of $HNO_3$, followed by the ionisation (or dissociation) and then progressive solvation at the surface leading to a partial solvation shell (Buch et al., 2002; Bianco et al., 2007, 2008). In a second stage, thought to be much slower, the adsorbed nitrate anions sink into the innermost crystal layers, leading to a complete solvation shell, and diffuse towards the bulk crystal. Recent studies addressed the ionisation state of $HNO_3$ adsorbed on ice surface, either using surface sensitive spectroscopy techniques (Křepelová et al., 2010; Marchand et al., 2012; Marcotte et al., 2013, 2015) or through molecular dynamics models (Riikonen et al., 2013, 2014). Molecular adsorbed state is found to be metastable, which happens only at very low temperatures (45 K), whilst ionic dissociation occurs irreversibly upon heating at 120 K (Marchand et al., 2012). Molecular dynamics simulations suggest a pico and subpicosecond ionisation of $HNO_3$ in the defects sites (Riikonen et al., 2013), further supporting that molecular adsorption of $HNO_3$ on ice is a fleeting state prior to ionisation, at least for environmentaly relevant temperatures.

Despite these recent improvements in the understanding of $HNO_3$ ionisation following adsorption on an ice surface, the transition between surface (adsorption) and bulk (diffusion) processes still needs to be fully characterised. To the best of our knowledge, no process-scale parameterisation





of the dissociation/solvation exists at the moment. Such parameterisation would be necessary to link surface and bulk concentrations, and further studies are thus needed to fully characterise the transition between these states. For this reason, both processes were treated separately in our model. The model configuration 1 (adsorption) is described in the next section, while the configuration 2 (solid state diffusion) is described in Sect. 3.3.

### 3.2 Model configuration 1: adsorption

The $HNO_3$ surface coverage is a function of temperature and pressure only. Crowley et al. (2010) presented a compilation of data evaluated by a IUPAC subcommittee, that characterises heterogeneous processes on the surface of solid particles, including ice. They recommend the use of a single-site Langmuir isotherm which gives the fractional surface coverage $\theta$:

$$\theta = \frac{N}{N_{\max}} = \frac{K_{\mathrm{LangP}} \, P_{HNO_3}}{1 + K_{\mathrm{LangP}} \, P_{HNO_3}} \qquad (2)$$

where $N_{\max} = 2.7 \times 10^{18}$ molecules m$^{-2}$ is the $HNO_3$ surface coverage at saturation,

$$K_{\mathrm{LangP}} = \frac{K_{\mathrm{LinC}} \, \mathcal{N}_A}{N_{\max} \, R \, T} \text{ (in Pa}^{-1}) \qquad (3)$$

$$K_{\mathrm{LinC}} = 7.5 \times 10^{-7} \exp\left(\frac{4585}{T}\right) \text{ (in m)} \qquad (4)$$

$K_{\mathrm{LangP}}$ and $K_{\mathrm{LinC}}$ are partition coefficients expressed in different units, $N$ is the $HNO_3$ surface
coverage (in molecules m$^{-2}$), $P_{HNO_3}$ is the $HNO_3$ partial pressure (in Pa), $\mathcal{N}_A$ is the Avogadro constant, $T$ the snow temperature (in K) and $R$ the molar gas constant ($R = 8.314$ J K$^{-1}$ mol$^{-1}$).

This parameterisation is established for temperatures ranging from 214 K to 240 K, and is used here at DC temperatures, typically in the 200–250 K range. The conversion of surface coverage to bulk concentration is done using the SSA value:

$$[HNO_3] = \frac{N \times SSA}{N_A} \qquad (5)$$

where $[HNO_3]$ is the nitrate concentration (in mol m$^{-3}$).

The results and discussion following adsorption calculation are presented in Sect. 4.

### 3.3 Model configuration 2: solid state diffusion

In configuration 2, the model computes solid state diffusion in a layered snow grain. Three dis-
tinct boundary conditions (BC) were successively used. Firstly, the $NO_3^-$ concentration at the air – ice interface was set according to thermodynamic equilibrium (BC1). In a second stage, the kinetic, co-condensation process was taken into account through an empirical, diagnostic parameterisation (BC2), then with a physically based prognostic parameterisation (BC3). The general diffusion scheme and specific BCs are presented in the next sections.





### 3.3.1 Diffusion scheme and equilibrium boundary condition (BC1)

Our model considers a spherical snow grain with a radius $R = 85$ µm, divided in concentric layers of constant thickness $\delta r = 0.05$ µm. In a first attempt labelled BC1, the outermost layer concentration (boundary condition of the diffusion equation) was set according to the thermodynamic equilibrium solubility of $HNO_3$ in solid solution as measured by Thibert and Dominé (1998):

$$X^0_{HNO_3} = 2.37 \times 10^{-12} \exp\left(\frac{3532.2}{T}\right) P^{1/2.3}_{HNO_3} \tag{6}$$

where $X^0_{HNO_3}$ is the molar fraction of $HNO_3$ in ice, $T$ is the snow temperature (in K) and $P_{HNO_3}$ is the $HNO_3$ partial pressure (in Pa).

The model then computes the solid state diffusion equation in spherical geometry with radial symmetry in the snow grain:

$$\frac{\partial C(r,t)}{\partial t} = D\left(\frac{2}{r}\frac{\partial C(r,t)}{\partial r} + \frac{\partial^2 C(r,t)}{\partial r^2}\right) \tag{7}$$

where $C(r,t)$ is nitrate concentration in the layer of radius $r$ at time $t$, and $D$ is the diffusion coefficient of $HNO_3$ in ice provided by Thibert and Dominé (1998):

$$D = 1.37 \times 10^{-2610/T} \text{ (in cm}^2\,\text{s}^{-1}) \tag{8}$$

Thibert and Dominé (1998) indicated uncertainties of $\pm 20$ % for equilibrium solubility, and of $\pm 60$ % for the diffusion coefficient, further explaining that the reported diffusion coefficient is probably the upper limit because of the existence of diffusion short pathways. The study of Thibert and Dominé (1998) was carried out at temperatures ranging from -8 °C to -35 °C. Nevertheless, Eq. (6) and (8) are applied to the temperature of DC surface snow, potentially leading to increased uncertainties.

The results and discussion of the modelling of nitrate concentration in surface snow using this BC1 approach are presented in Sect. 5.1. We also investigated how the uncertainties over the solubility and the diffusion coefficient affect the simulations, in a sensitivity study presented in Sect. 5.4.

### 3.3.2 Diagnostic co-condensation parameterisation (BC2)

An empirical, diagnostic parameterisation of the co-condensation process was firstly developed to investigate the concentration of the growing phase. Valdez et al. (1989) carried out experiments on $SO_2$ incorporation into ice growing from the water vapour, and reported that the amount of sulfur incorporated into the ice increased linearly with the amount of ice deposited. Jacob and Klockow (1993) compared the concentration of $H_2O_2$ in the gas phase and in the snow during fog events, and showed that the molar fraction of hydrogen peroxide, $X_{H_2O_2}$, resulting from co-condensation was similar to the ratio of partial pressures: $X_{H_2O_2} \simeq \frac{P_{H_2O_2}}{P_{H_2O}}$, as previously hypothesised by Sigg and Neftel (1988). Dominé et al. (1995) refined this analysis using the kinetics theory of gases to include





the number of collisions, and further taking into account the surface accommodation coefficients $\alpha$. They proposed that the molar fraction of a gas $i$ ($X_i$) condensating along with water vapour should obey to the following equation, where $M$ is the molar mass:

$$X_i = \frac{P_i}{P_{H_2O}} \frac{\alpha_i}{\alpha_{H_2O}} \sqrt{\frac{M_{H_2O}}{M_i}} \qquad (9)$$


However, Ullerstam and Abbatt (2005) carried out laboratory measurements of $HNO_3$ concentration in growing ice, and their results suggested that $HNO_3$ concentration was proportional to $P_{HNO_3}^{0.56}$ and independent of the water vapour partial pressure:

$$\log_{10}(X_{HNO_3}) = 0.56 \times \log_{10}(P_{HNO_3}) - 3.2 \qquad (10)$$

where the exponent 0.56 could be explained by acid dissociation during co-condensation. Another 335 possible explanation proposed by Ullerstam and Abbatt (2005) is that thermodynamic solubility governs at least partially the composition of a growing crystal as $HNO_3$ is sufficiently volatile and mobile to be excluded from the growing ice. Indeed, the power 0.56 dependence to $HNO_3$ partial pressure is close to that of thermodynamic equilibrium solubility (in Eq. (6), $1/2.3 \simeq 0.43$).

To summarise the conclusions of these studies, the co-condensed phase has a concentration which 340 depends on (i) the studied trace gas partial pressure (but without agreement on the exponent in the case of $HNO_3$) and (ii) may or may not depend on the water vapour partial pressure. Thus, in order to test these hypotheses, a first simple diagnostic parameterisation of co-condensation process was implemented by adding an adjustable term in the boundary condition definition of the concentration:


$$X_{HNO_3} = X_{HNO_3}^0 + \alpha \cdot P_{HNO_3}^\beta \cdot P_{H_2O}^\gamma \qquad (11)$$

where $X_{HNO_3}^0$ is the molar fraction of $HNO_3$ in ice given by thermodynamic equilibrium (see Eq. 6), $P_{HNO_3}$ and $P_{H_2O}$ are partial pressures of $HNO_3$ and water vapour, respectively (in Pa), and $\alpha$, $\beta$, and $\gamma$ are adjusted parameters. The results of this BC2 configuration are presented in Sect. 5.2.

### 3.3.3 Prognostic co-condensation parameterisation (BC3)

In order to develop a physically based, prognostic parameterisation of the co-condensation process (BC3), the second step was to define the growth rate of snow crystals submitted to a temperature gradient. Calculation of the water vapour gradient inside the snowpack is a complex matter (Flin and Brzoska, 2008). Using upscaling theories, several recent studies aimed at obtaining macroscopic 355 parameterisations ensued from an accurate description of the processes (heat conduction, vapour diffusion, sublimation and condensation) occurring at the microscopic scale (Miller and Adams, 2009; Pinzer et al., 2012; Calonne et al., 2014; Hansen and Foslien, 2015). A major issue may arises when simply upscaling microscopic laws by using averaged, macroscopic parameters such as the temperature gradient. Indeed, as illustrated by Calonne et al. (2014, Fig. 4), microscale inhomogeneities





are likely to enhance locally the temperature gradient, and thus the flux of water vapour. However, Pinzer et al. (2012) compared the mass flux calculated using a macroscopic diffusion law on the one hand, and using two microscopic computations (particle image velocimetry and finite element simulation) on the other hand. They concluded that "the three methods of calculation coincide reasonably well", and thus that "the macroscopic vapour flux in snow can be calculated once the temperature

gradient and the mean temperature of the snow are known, independently of the microstructure". In the macroscopic diffusion law equation, Pinzer et al. (2012, Eq. (3)) used an effective diffusion coefficient for water vapour in the interstitial air, whose value has been a subject of debate for a long time (Calonne et al., 2014, and ref. therein). In their study, Calonne et al. (2014) concluded that the effective vapour diffusion is not enhanced in snow.

Based on these results, we assumed that a macroscopic scale water vapour flux can be reasonably estimated using macroscopic, mean parameters. Following particulate growth laws in cloud models, Flanner and Zender (2006) proposed an equation giving the mass variation over time as a function of the water vapour gradient:

$$\frac{dm}{dt} = 4\pi R^2 D_v \left( \frac{d\rho_v}{dx} \right)_{x=R} \tag{12}$$

where $R$ is the particle radius, $D_v$ is the diffusivity of water vapour in air, and $\rho_v$ is the water vapour density (in $\mathrm{kg\,m^{-3}}$). The diffusivity of water vapour in air can be found in Pruppacher and Klett (1997) as a function of pressure and temperature, in the -40 °C to +40 °C range:

$$D_v = 2.11 \times 10^{-5} \left( \frac{T}{T_0} \right)^{1.94} \frac{P_0}{P} \text{ (in } \mathrm{m^2\,s^{-1})} \tag{13}$$

where $T_0 = 273.15$ K and $P_0 = 101325$ Pa. We stress here that the water vapour gradient in Eq.

(12) is originally intended to be a microscopic gradient, but a macroscopic gradient derived from the modelled temperature profile in the two uppermost layers was used. Because this growth law is used to parameterise the co-condensation process, only the cases leading to a mass increase were taken into account. Finally, the mass growth rate defined by Eq. (12) can be converted into a volume growth rate using ice density $\rho_{\mathrm{ice}}$, and then to a radius growth $\Delta r$ (in m) by assuming a uniform

condensation on the whole grain surface during a time $\Delta t$:

$$\Delta r = \sqrt[3]{\frac{3}{4\pi} \left( \frac{1}{\rho_{\mathrm{ice}}} 4\pi R^2 D_v \left( \frac{\Delta \rho_v}{\Delta x} \right)_{x=R} \Delta t \right) + R^3} - R \tag{14}$$

Note that in this equation $\Delta r$ depends on $\Delta t^{1/3}$.

The last step of co-condensation parameterisation is to implement this dynamic feature of a growing crystal into the fixed shape of a spherical grain. An accurate modelling of temperature gradient

metamorphism and ensuing co-condensation process would require a complex description of the system, including snow grain shape, direction of growth, and local inhomogeneities, which is within the purview of snow microphysics 2-D or even 3-D state of the art models (see for example Flin et al., 2003; Kaempfer and Plapp, 2009; Calonne et al., 2014).





Another difficulty comes from the competition between co-condensation and diffusion. It was observed that the co-condensation process leads to out of thermodynamic equilibrium concentrations (Bales et al., 1995; Dominé and Thibert, 1995, 1996; Ullerstam and Abbatt, 2005) that enhance solid state diffusion to re-equilibrate. The combination of these two processes was studied by Dominé and Thibert (1996) who proposed a theoretical description through a two-stages process. Firstly, a layer of given thickness ($h$) and composition ($X_{\mathrm{kin}}$) condensates at $t = 0$. Then, solid state diffusion takes place to re-equilibrate this layer towards the equilibrium concentration ($X_{\mathrm{eq}}$), until another layer condensates at $t = \tau$, isolating the previous layer. According to this simplified description, the resulting molar fraction at a distance $d$ from the surface and after a diffusion time $t$ is given by:

$$X(d,t) = X_{\mathrm{kin}} + (X_{\mathrm{eq}} - X_{\mathrm{kin}}) \, \mathrm{erfc} \left( \frac{d}{2\sqrt{D\,t}} \right) \tag{15}$$

where $X_{\mathrm{kin}}$ is the molar fraction of the growing phase (which could be provided either by the gas kinetics theory parameterisation, Eq. (9), or by the empirical relation, Eq. (10)), $X_{\mathrm{eq}}$ is the molar fraction inferred from thermodynamic equilibrium solubility (Eq. 6) and $D$ is the diffusion coefficient of $HNO_3$ in ice (Eq. 8).

In Eq. (15), $\mathrm{erfc}$ is the complementary error function, with $\mathrm{erfc}\,(0) = 1$ and $\mathrm{erfc}\,(x)$ is decreasing towards zero for positive values. Since $\sqrt{D\,t}$ represents the typical diffusion length during a time $t$, the resulting molar fraction given by Eq. (15) will be close to $X_{\mathrm{eq}}$ if the condensed layer is thin compared to the typical diffusion length, i.e. if the layer readily re-equilibrates through diffusion. On the contrary, if the condensed layer is thick, the resulting molar fraction gets closer to $X_{\mathrm{kin}}$.

Following Dominé and Thibert (1996), the BC3 boundary condition defining the outermost layer concentration is set as $X(\Delta r, \Delta t)$ (Eq. 15) where $\Delta r$ is the thickness of the condensed layer which has grown during the timestep $\Delta t$ (Eq. 14).

## 4 Results and discussions for model configuration 1

The simulated nitrate concentration of the snow skin layer obtained in model configuration 1, involving only the adsorption process, is presented and discussed in this section.

### 4.1 Results

The evolution of the concentration of nitrate in the snow skin layer is plotted in Fig. 3. Undeniably, the adsorbed concentration modelled using non-dissociative Langmuir isotherms parameterisation does not fit with the measured concentration in three ways: firstly, the modelled concentration is higher than the measured ones during most of the year. From February to August, the modelled concentration is 2.5-fold higher than the measured one, and this ratio increases to 8.3 from September to mid-November (see vertical separations in Fig. 3). On the contrary, the modelled concentration gradually decreases till end January while the measured one reaches a seasonal maxima, leading to a ratio of 0.62 between modelled and measured concentrations during this last period. Secondly,





the modelled concentration shows a strong diurnal variability following temperatures, with a ratio
between daily maximum and minimum concentration regularly higher than 5, and with a yearly
average equal to 2.6. By contrast, field measurements show weak diurnal variations of nitrate con-
centration in surface snow, and no anticorrelation with temperature (Fig. 2 in Supplement). The third
major discrepancy is a premature seasonal maximum in the computation, starting late August and
reaching maximum early November, while concentration measured in snow lags by 65 days.

The features of the modelled concentration attributed to adsorbed nitrate can be explained by
the temperature and partial pressure dependencies of the adsorption isotherm. The surface coverage
parameterisation strongly decreases with temperature (exponential function of the reciprocal tem-
perature in Eq. (4)), whilst it increases roughly linearly with the $HNO_3$ partial pressure when the
surface coverage is well below saturation. This explains the strong diurnal variations following the
temperature cycle. It also explains the yearly pattern of the modelled concentration: firstly, during
the winter, the very low temperature prevails over the low $HNO_3$ partial pressures, leading to mod-
elled concentration already much higher than that measured. The influence of temperature is easily
seen in April, May, and August, when temperature is the lowest (see Fig. 1b), leading to higher
modelled concentration than in June and July, when temperature is higher and $HNO_3$ partial pres-
sure is alike. Then, from early September to early November, $HNO_3$ partial pressure increases while
temperature shows only a moderate increase, leading to the modelled peak of absorbed nitrate. Fi-
nally, nitrate partial pressure stays high until January, but this is counterbalanced by the temperature
which increases to its yearly maximum, compelling modelled surface coverage to fall well under the
measured values.

### 4.2 Discussion

Despite the use of the IUPAC current recommendation for the parameterisation of $HNO_3$ adsorp-
tion on ice, the modelled quantities adsorbed on snow are clearly incompatible with the measured
concentration. In order to explain this discrepancy, we compared the experimental setups used in the
various studies of adsorption (Abbatt, 1997; Arora et al., 1999; Hanson, 1992; Hudson et al., 2002;
Hynes et al., 2002; Laird and Sommerfeld, 1995; Leu, 1988; Sokolov and Abbatt, 2002; Ullerstam
et al., 2005; Zondlo et al., 1997). A review of these studies, and of the experimental techniques
used, can be found in (Huthwelker et al., 2006). In brief, two main experimental techniques prevail:
flow tubes, which were mostly used (Abbatt, 1997; Arora et al., 1999; Hanson, 1992; Hynes et al.,
2002; Leu, 1988; Sokolov and Abbatt, 2002; Ullerstam et al., 2005), and Knudsen cells, which were
used in two studies (Hudson et al., 2002; Zondlo et al., 1997). Whatever the technique used, ice was
deposited on the reactor walls either by water vapour condensation (Hanson, 1992; Hudson et al.,
2002; Leu, 1988; Zondlo et al., 1997), or by fast freezing an ice film (Abbatt, 1997; Hynes et al.,
2002; Sokolov and Abbatt, 2002; Ullerstam et al., 2005).



A first pitfall which may arise from these studies come from the lack of quantification of the exposed surface area of ice, which was measured only once by Hudson et al. (2002). They carried out several experiments at 209, 213 and 220 K, and found that the exposed surface was twice the geometrical surface. Leu et al. (1997) found that this ratio can be as high as $\sim 9$ in the case of ice formed by water vapour deposition at 196 K. These authors also reported that this ratio increases with the amount of water deposited, and also increases when the temperature decreases. On the other hand, in another study using ice formed by fast freezing a film of water, Abbatt et al. (2008) concluded that the ice was smooth at a molecular level, implying a ratio near 1. Yet, except in the study of Hudson et al. (2002), an under-estimation of the exposed surface, which leads to an overestimation of the surface coverage of ice, can not be ruled out.

All adsorption studies assumed that at very low temperatures, diffusion in bulk ice is negligible. However, even if the fraction of $HNO_3$ entering the bulk ice is small, neglecting it leads to a systematic overestimation of the surface coverage. Cox et al. (2005) analysed the data of Ullerstam et al. (2005) to include the diffusion process. Their study brought new insights about surface versus bulk processes, and their model performed well in reproducing adsorption curves when diffusion into the bulk was also taken into account. However, instead of using the existing parameterisation for nitrate solubility and diffusion coefficient in the ice (see Sect. 3.3.1), they made use of a simplified scheme to consider the diffusion process, which includes an adjustable rate coefficient for diffusion and hinder a close comparison with the above mentioned paramterisations. Furthermore, the desorption curves could not be well fitted by their model, especially for low surface coverage, indicating that the involved processes are still not fully understood and constrained.

The diffusion of nitrate into bulk ice could also have been further enhanced for three distinct reasons. Firstly, it is noteworthy that if the exposed surface area of ice is larger than the geometric surface, this leads to a larger exchange interface, thus increasing the amount of $HNO_3$ diffusing to bulk ice in the total uptake. On the other hand, even if the ice covering the reactor's walls was smooth in the case of a frozen liquid film, the fast freezing process very likely leads to a highly polycrystalline structure, where grain boundaries may act as shortcuts for the diffusion, thus enhancing bulk uptake. Last, several authors (Hudson et al., 2002; Hynes et al., 2002) pointed out that despite the careful attention to ensure that ice surface was in equilibrium with its vapour, if the exposed ice was slightly growing because of slight supersaturation or due to the highly dynamic air–ice interface (Bolton and Pettersson, 2000), part of the observed uptake could be ascribed to bulk incorporation of $HNO_3$ with condensing water.

More generally, the question of the adsorbed state, closely linked to the ionisation process and to the reversibility of the adsorption, can also contribute to explain the mismatch between current parameterisation and measurements. In all the uptake experiments, it was observed that the total uptake splits between reversible and irreversible components, the former being only a minor part of the total. For instance, Ullerstam et al. (2005) reported that on average 20 % of the initial uptake





was desorbing. Should a part of this irreversible uptake already account for a strongly bound, bulk uptake, that could explain a major part of the overestimation of the modelled absorbed concentration. New investigations are needed to gain a clearer view of the partitioning between surface and bulk.

Finally, several other uncertainties can be invoked to explain the discrepancies. The saturated surface coverages reported in the various studies range over almost one order of magnitude, from
$1.2 \times 10^{14}$ molec cm$^2$ (Arora et al., 1999) to $1.0 \times 10^{15}$ molec cm$^2$ (Hynes et al., 2002). This uncertainty directly impacts the modelled surface coverage (Eq. 2). Secondly, most adsorption studies used HNO$_3$ partial pressure between 2 and 3 orders of magnitude higher than the one relevant at DC. Ullerstam et al. (2005) improved this, by using partial pressures down to $\sim 9 \times 10^{-7}$ Pa, however this remains $\sim 25$-times higher than the lowest partial pressures measured in winter at DC
($\sim 3.5 \times 10^{-8}$ Pa). Using their parameterisation in DC conditions thus implies a great extrapolation. The lack of data for very low partial pressures also enhances the uncertainties over the relevant type of adsorption isotherms, as the behaviour in the unsaturated region (i.e. at low partial pressure) provides more constraint over the best type of adsorption isotherms than that in (or near) the saturated region. This explains why several kind of isotherms (dissociative (Hynes et al., 2002) or
non-dissociative Langmuir isotherm (Ullerstam et al., 2005), Frenkel-Halsey-Hill isotherm (Hudson et al., 2002)) have been proposed, but no clear consensus has been achieved.

In order to test these different explanations, experimental setups should systematically include measurements of the exposed area of ice, and use partial pressures as low as possible. Processing the raw experimental data with the approach developed by Cox et al. (2005) seems a promising way
to discriminate between surface and bulk uptake processes. Improvements of this approach could probably be achieved by using state-of-the-art parameterisation of the diffusion process.

Regarding the present study uncertainties, snow temperature and SSA and HNO$_3$ partial pressure are the three variables controling the adsorbed surface coverage. HNO$_3$ partial pressure, assumed to be equal to the total atmospheric nitrate (see Sect. 2.1.1), is thus the upper limit. However, as pre-
sented in the data description (see Sect. 2.1.1), this assumption likely leads to an overestimation no larger than 20 % on average, which cannot explain the overestimation of the modelled concentration by a factor of 2.5–8.3. On the contrary, the warm bias of modelled temperatures (see Sect. 2.2.1 and SI 1) leads to smaller modelled adsorption concentration, and the slightly reduced diurnal amplitude tends to reduce this other discrepancy between modelled and measured concentration. Last, the SSA
was kept constant during the whole simulation, but a recent study by Libois et al. (2015) indicated that the SSA value is relevant to summertime but is 2-3 lower than the wintertime SSA observations (see Sect. 2.2.2). At that time of the year, the modelled adsorbed concentration is already highly overestimated, thus accounting for a higher SSA would increase the discrepancy.

To conclude this section, several reasons were invoked to explain the overestimation of the mod-
elled adsorbed concentration. Given the inability of the current parameterisation to fit the measurements and the major uncertainties related to the adsorption process, we decided thereafter to ignore




the adsorbed concentration. In order to estimate the error thereby induced, we make the rough hypothesis that the current adsorption parameterisation is flawed by a constant overestimation factor. Decreasing the modelled adsorbed concentration by a constant factor of $\sim 20$ so that its enveloppe

never exceed measured concentration leads to small adsorbed concentration during most of the year excepted in early spring, i.e. in the September – early November peak period (see Fig. 3). In this situation, we estimate that adsorbed nitrate accounts for less than 13 % of snow nitrate on yearly average (less than 9 % when excluding the early September to early November period, and almost 30 % during these 2 months), thus neglecting the adsorption process should lead to only minor er-

ror, except during spring. One way to test this hypothesis is to carry out hourly measurements of nitrate concentration in surface snow during spring. Owing to the strong temperature dependency of the adsorption isotherm, if adsorbed nitrate accounts for an important fraction of snow nitrate, then significant daily variations of snow nitrate concentrations should be observed.

## 5   Results and discussions for model configuration 2

In this section, the model was run in configuration 2, based on the solid state diffusion process (see Sect. 3.3). The results obtained with the three distinct BC parameterisations are successively presented and discussed hereafter.

### 5.1   Thermodynamic equilibrium concentration (BC1)

The first attempt to model nitrate concentration in the skin layer was done using solely the thermo-
dynamic equilibrium concentration (see Sect. 3.3.1 and Eq. 6) to constrain the concentration of the external layer of the snow grain (BC1). The resulting concentration is plotted in Fig. 4 along with the measured concentration. The initial value of $\sim 500$ ng g$^{-1}$ and the sharp decrease at the beginning of the serie (30/01/09 – 07/02/09) are due to the initialisation of the whole grain concentration to the closest measurement (point not shown, a few hours before the start of the simulation) and should not
be interpreted.

From mid April to late October, the modelled concentration is in reasonable agreement with the measured concentration, with some features appearing to be reproduced by the model (a slight, steady increase lasting from July to August, followed by a trough and then a second slight increase period from September to mid-October). During this winter period, modelled concentration appears
to be often slightly lower than measurements, and that point will be further discussed in the sensitivity study presented in Sect. 5.4. The modelled concentration also features smoother variations than the measured concentration, which can be mainly explained by the time resolution of HNO$_3$ partial pressure used as input, of roughly one week (see Sect. 2.1.1 and Fig. 1a). The good consistency between modelled and measured concentrations during winter months is an important result,



as this indicates that wintertime concentration of nitrate in surface snow is mainly driven by the
thermodynamic equilibrium solubility, coupled to solid state diffusion.

On the other hand, this first modelling attempt clearly fails to reproduce the summer peak of ni-
trate concentration in snow, with values in the 50-200 ng g$^{-1}$ range from November to early April,
while measured concentration peaks above 1400 ng g$^{-1}$. These results also show that summertime

concentration of nitrate in surface snow is highly enriched compared to what is expected from the
thermodynamic equilibrium. These results demonstrate that another uptake process, driven by kinet-
ics rather than thermodynamics, is needed to explain such high summertime concentration.

### 5.2   Diagnostic co-condensation parameterisation (BC2)

The BC2 includes the kinetic co-condensation process, through the empirical diagnostic parame-

terisation presented in Sect. 3.3.2. We adjusted the 3 coefficients in Eq. (11) in order to minimise
the RMSE between modelled and measured snow nitrate concentration. The optimal result, plot-
ted in Fig. 5, was obtained with $X_{HNO_3} = X_{HNO_3}^0 + \alpha \cdot P_{HNO_3}^{0.43} \cdot P_{H_2O}^{1.27}$. The $\alpha$ parameter value was
adjusted so that the amplitude of the modelled summer peak fit the data, but has no physical signifi-
cation. However, the most relevant point to note is that the modelled peak is well in phase with the

measurements (as a main difference with the adsorption), and features similar shape. Furthermore, it
is noteworthy that including the co-condensation has not degraded the wintertime prediction. Indeed,
because of the very low winter temperature at DC, and given the exponential dependency of water
vapour pressure over temperature, the co-condensation term becomes almost negligible (Town et al.,
2008).

The optimum exponent for HNO$_3$ partial pressure is $0.43$ which exactly corresponds to the ex-
ponent for HNO$_3$ partial pressure of thermodynamic equilibrium concentration (in Eq. (6), $1/2.3 \simeq$
$0.43$). Even if that needs to be confirmed by further investigations, this result tends to confirm the
hypothesis formulated by Ullerstam and Abbatt (2005) that thermodynamic partitioning plays a role
in the co-condensation process (see Sect 3.3.2).

Because of the correct timing and shape of the modelled peak of nitrate, these results suggest that
the co-condensation process is responsible of the out of equilibrium, high concentration of nitrate
in the skin layer in summer. Among the two available laws giving $X_{kin}$, the concentration of the
co-condensed phase (see Sect. 3.3.2 , Eq. (9) or (10)), the empirical one, whose dependency over the
HNO$_3$ partial pressure is the closest to $0.43$, seems the more suited to reproduce the observations.

### 5.3   Prognostic co-condensation parameterisation (BC3)


The last part of this work aimed at refining the parameterisation for the co-condensation process,
using physically based variables. The prognostic parameterisation developed hereafter is refered as
BC3. For sake of simplicity, and because the growth of snow grain is very slow compared to the
recycling of vapor as suggested by Pinzer et al. (2012), a constant radius ($R$) is assumed. However,





the growth law defined in Eq. (12) is used in order to evaluate the equivalent radius increase $\Delta r$ resulting from the co-condensation process during the model timestep $\Delta t$ (Eq. 14). Finally, the concentration resulting from concomitant thermodynamic process (diffusion equilibration) and kinetic process (co-condensation process) is calculated using the theoretical Eq. (15) at a depth $\Delta r$, that is at the surface of the modelled snow grain whose radius is supposed constant.

The resulting modelled nitrate concentration in surface snow is presented in Fig. 5. In Table 1, a summary of the model runs, along with their RMSE, is presented . Simulation results are similar to those obtained with the BC2 parameterisation, but with a slightly improved RMSE. In this physically based parameterisation, however, a slight dependency of the results to the model timestep arises. This is explained by the radius increase $\Delta r$ which depends on the cubic root of the time (Eq. 14), and

which is divided by the square root of the time in Eq. (15). To compensate this dependency, either the timestep of the model needs to be adjusted for optimum results, or an additional correction factor can be used in order to keep the timestep unchanged, with a value well suited regarding the diffusion process. The exact reason of this dependency over the time step is complex to establish, but can very likely be ascribed to the hypothesised geometry of the snow grain (a sphere) and of the condensed

phase (a layer). Improving this point necessitates determination of the relationship between mean thickness of the co-condensed layer as a function of time.

In Fig. 5, the modelled concentration shows a poorer fit with the measured concentration during spring, just before the observed peak of snow nitrate. This is confirmed by a monthly regression analysis (see Table SI 1) which shows a lower correlation from September to November, which

corresponds to the period where the modelled adsorption peaks (see Fig. 3). This is another clue to say that adsorbed nitrate may account for a noticeable part of surface snow nitrate in early spring.

Given the numerous assumptions made in the model, the overall reproduction of the measurements by the parameterisation including co-condensation appears satisfactory.

### 5.4 Sensitivity study

In order to further investigate the modelling uncertainties, the sensitivity of the model to the thermodynamic equilibrium concentration, diffusion coefficient and SSA value is evaluated. A synthesis of RMSE values of the sensitivity runs is presented in Table 1.

As shown in Sect. 5.1, wintertime modelled concentration underestimates the measurements, which could be explained by an underestimated thermodynamic equilibrium solubility (Eq. 6). The

best fit with the data is obtained for an increase of 39 % (see Table 1). This optimum increase is almost twice as much as the uncertainty reported by Thibert and Dominé (1998, 20 %), however we applied the solubility parameterisation at much lower temperature than in their study.

A few measurements of the ratio of $HNO_3$ over atmospheric nitrate presented in Sect. 2.1.1 suggest that $HNO_3$ might account for roughly 70–90 % of atmospheric nitrate. Taking this ratio into

account would reduce the $HNO_3$ partial pressure used as input in the model, but might be counter-



balanced by a further increase of the thermodynamic solubility. New studies are needed to confirm the speciation of atmospheric nitrate and its seasonal variation. On the other hand, the current underestimation of the modelled concentration during wintertime can also be partly ascribed to a small adsorbed fraction amongst the total snow nitrate.

Secondly, using a diffusion coefficient lower than that suggested by Thibert and Dominé (1998, Eq. 8) generally improves the simulation performance. Using BC3 simulation as a reference, decreasing the diffusion coefficient by 72 % leads to the best reproduction of the results (see Table 1). When the solubility value increased by 39 % is used, the diffusion coefficient is decreased by 64 %. Thibert and Dominé (1998) reported a 60 % uncertainty for the diffusion coefficient, and indicated

that their parameterisation likely represents the upper bounds, which compares well with the sensitivity analysis result.

However, another explanation is possible, because a decrease of the SSA has a similar effect to a decrease of the diffusion coefficient, as they both slow down the diffusion. Decreasing the SSA to 23 $m^2\,kg^{-1}$ leads to almost the same result as a decrease of 64 % of the diffusion coefficient (see

Table 1). In the current version of the model, the radius of the snow grain is kept constant over time as a simple hypothesis, but it has been shown by Picard et al. (2012); Libois et al. (2015) that snow grain size features a sharp increase at DC during December and January, when the modelled water vapour fluxes driving the co-condensation process are highest. It is remarkable that the optimum value of 23 $m^2\,kg^{-1}$ is in very good agreement with that observed in summer Libois et al. (2015,

Fig. 1). Future development of the current work should consider grain size change to distinguish between these two alternative hypotheses.

## 6    Conclusions

In this study we investigated the role of three processes that intervene in air–snow exchange of nitrate at DC, which revealed that the co-condensation of nitrate along with the condensation of

water vapour flux driven by thermal gradient metamorphism is a major process, absolutely needed to explain the summer peak of nitrate measured in surface snow.

This study further reveals that the current state-of-the-art parameterisation for $HNO_3$ adsorption on snow leads to modelled concentration which differs from the observations, and cannot be used without major changes. We propose the hypothesis that adsorption measurements of $HNO_3$ on ice

attributed most, if not all, of the uptake to the only adsorption process, while a noticeable part of this uptake should in fact be ascribed to bulk, irreversible incorporation. New laboratory investigations should probably be conducted along with theoretical studies in order to improve the current understanding of the binding process occurring on the ice surface and its kinetics, in order to make a clearer distinction between surface and bulk nitrate on the ice. On the contrary, studies aiming at the

determination of equilibrium solubility and diffusion coefficient of nitrate in the ice take advantage



of "integrative" measurements, in the sense that these two properties are deduced from macroscopic concentration profiles in the ice, without needing further hypothesis or insight about the actual microscopic processes occurring at the air–ice interface (binding, ionisation, solvation). This different approach probably explains why, despite being much less numerous, these studies provided robust
parameterisations.

Thus, by getting rid of the adsorption process, and focusing solely on the solid state diffusion inside a spherical snow grain, we developed a physically based parameterisation for the concentration at the surface of this grain, used as boundary condition of the diffusion equation. This parameterisation includes both thermodynamic equilibrium concentration and co-condensation process. Without
needing any further adjustment parameter, the implementation of this newly developed parameterisation allowed a satisfactory reproduction of the one-year long dataset of nitrate concentration in DC surface snow. Given the resemblance in the general features of the measurements of atmospheric and snow nitrate in other Antarctica sites such as South Pole or even Halley, it seems very likely that the overall modelling framework that we developed can generalise at least over the Antarctic plateau.

Even if some improvements still need to be done, especially regarding a more realistic geometry of the co-condensed phase, the developed parameterisation and the overall modelling scheme can already be implemented as a foundation piece in 1-D, snow–atmosphere models. Some new insights over nitrogen recycling inside the snowpack could ensue from such vertical, 1-D modelling.

Ultimately, this work shows that snow physics and snow chemistry are tightly coupled, and espe-
cially that snow metamorphism resulting mainly from temperature gradients does not affect solely the physical properties of the snow, but also its chemical composition. It is also noteworthy that physical exchange processes on their own appear to explain a major part of the observed changes in surface snow nitrate at DC. Thus, it seems highly necessary that any field campaign mainly dedicated to snow chemistry also devote efforts to precise measurements of snow physical properties.

*Author contributions.* J

. Savarino initiated this study on the basis of field data collected in the framework of NITE DC program. J. Bock developed the model code and performed the simulations. G. Picard carried out the surface energy budget and thermal diffusion simulations to get the snow temperature. All co-author contributed to the development of the modelling framework. J. Bock prepared the manuscript with contributions from all co-authors.

*Acknowledgements.* We wish to thank Frédéric Flin for helpful discussions about water vapour exchange and its parameterisation inside the snowpack. We are grateful to Emmanuel Witrant and David Stevens for helpful discussions about the implementation of various boundary conditions of the diffusion equation. We thank James France and Max Thomas for proofreading the final manuscript. J. Bock is grateful to Christian George for co-supervising his PhD.





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





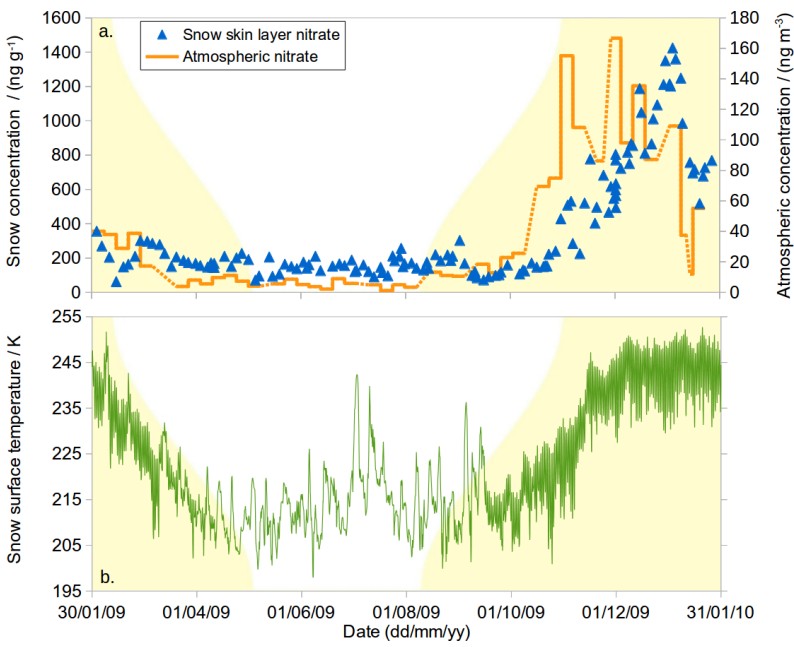

**Figure 1. (a)** Atmospheric nitrate concentration (orange lines, right axis) and snow skin layer nitrate concentration (blue triangles, left axis). **(b)** Modelled surface snow temperature. In both panels, the back yellow coloured area is proportional to sunlight duration.

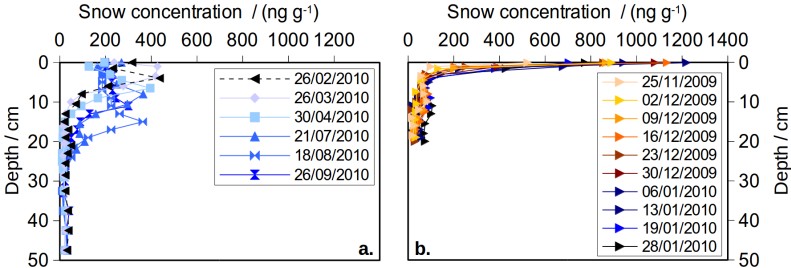

**Figure 2. (a)** Winter nitrate concentration profiles (in ng g$^{-1}$) in snow pits. **(b)** Summer nitrate concentration profiles (in ng g$^{-1}$). The measurement date (dd/mm/yyyy) is indicated.





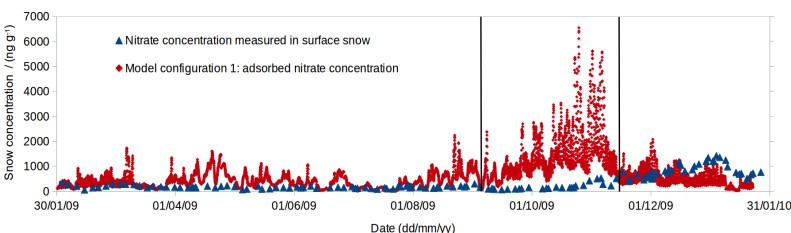

**Figure 3.** Concentration of nitrate in snow skin layer: measured concentration (blue triangles) and model configuration 1: adsorbed concentration (red diamonds). Note the y-axis scale change as compared to Fig. 1a. The output timestep is one hour. Verticals bars are visual aids to separate periods mentioned in the text.

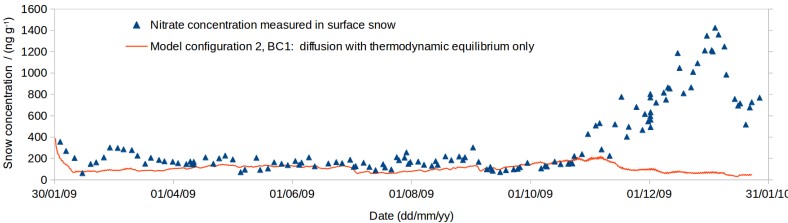

**Figure 4.** Nitrate concentration in the skin layer: measured concentrations (blue triangles) and model configuration 2 (orange line) using only thermodynamic solubility to constrain the air–snow partitioning (BC1). The output timestep is 4 hours.

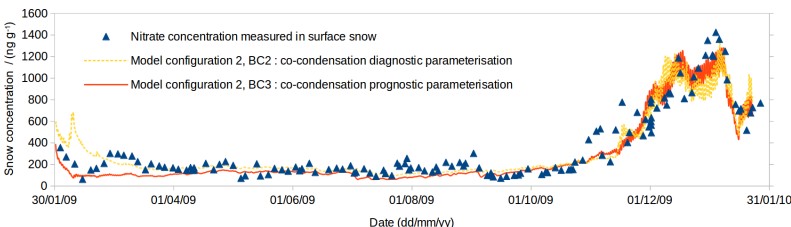

**Figure 5.** Nitrate concentration in the skin layer: measured concentrations (blue triangles) and mode configuration 2 using two distinct parameterisations of the co-condensation process: diagnostic parameterisation (BC2, dashed yellow line) and physically based prognostic parameterisation (BC3, solid red line).





**Table 1.** Summary of the main simulations with their description, along with the RMSE value to evaluate the discrepancy between modelled and measured values. If relevant, the numbering of the figure where results are plotted is indicated.

| Simulation description | RMSE / ng g$^{-1}$ | Fig. |
|---|---|---|
| Configuration 1: adsorption | 551 | 3 |
| Configuration 2: diffusion with thermodynamic solubility only (BC1) | 437 | 4 |
| Configuration 2: diffusion with diagnostic parametrisation of the co-condensation (BC2) | 124 | 5 |
| Configuration 2: diffusion with prognostic parameterisation of the co-condensation (BC3) | 116 | 5 |
| Sensitivity study, solubility increased by 39 % | 110 | |
| Sensitivity study, diffusion coefficient decreased by 72 % | 100 | |
| Sensitivity study, solubility increased by 39 % and diffusion coefficient decreased by 64 % | 96 | |
| Sensitivity study, solubility increased by 39 % and SSA value decreased to 23 m$^2$ kg$^{-1}$ (initial value = 38 m$^2$ kg$^{-1}$) | 96 | |