# Peer review of "Air – snow exchange of nitrate: a modelling approach to investigate physicochemical processes in surface snow at Dome C, Antarctica"

_Atmospheric Chemistry and Physics, 2016_

## Referee Comment (RC1) · Anonymous Referee #1 · 7 Apr 2016

Bock et al. use a model to examine the impacts of the physical exchange processes, adsorption, bulk diffusion and co-condensation, on the depth profiles of nitrate concentration in snow at Dome C, Antarctica. They find that bulk diffusion and co-condensation alone can explain the observed profiles.

This paper was not clearly written and thus very frustrating to read. I don't understand how one can model snow nitrate without including important processes such as atmospheric deposition and the photolysis of snow nitrate. The former is how nitrate gets to the snow in the first place, and the latter has been shown to be the dominant loss process of snow nitrate at Dome C. Although I agree that their study is important, as such physical processes will influence the distribution of nitrate in the snow column and

the snow grain, the latter of which may influence e.g., how photolabile snow nitrate is, I don't see how they can ignore these other important processes. It seems that it would be better to use their model to examine the results in a laboratory, where the processes they ignore can be controlled.

Since the manuscript is not clearly written, it is possible that I am misunderstanding something important about their modeling framework.

---

## Referee Comment (RC2) · Anonymous Referee #2 · 14 May 2016

My sincere apologies to the authors and the editor for the delayed submission of my referee comments.

[Summary]

In this paper, Bock et al. develop numerical models to analyze physical mechanisms behind the incorporation of nitrate from ambient air into surface snow at Dome C, Antarctica. Four different types of the models are developed, each of which is applied to simulate the evolution of nitrate concentrations in a single grain of the surface snow over one annual cycle. Each model is constrained by measured concentrations of total nitrate (gaseous + particulate, but assumed with some rationales to be predominated by the gaseous component) in ambient air, invariant grain size and specific

surface area assumed on the basis of previous field measurements, and simulated snow temperatures by using a 1-D snow physics model previously developed by one of the authors. The first model assumes that the uptake of nitrate occurs entirely via surface adsorption on the snow grain. The second model ("BC1 model") assumes that the solid-state diffusion and thermodynamic equilibrium solubility of HNO3 in bulk ice determines the uptake of HNO3 from ambient air. The third and fourth models ("BC2 model" and "BC3 model", respectively) assume that the co-condensation of HNO3 and H2O (associated especially with snow-grain metamorphism in the summer) augments the uptake of HNO3 from that predicted by the process of solid-state diffusion and thermodynamic equilibrium represented in the BC1 model. The difference between the BC2 and BC3 models is that the former parameterizes the co-condensation process in a diagnostic fashion where empirical parameters are adjusted to match the model with the measurements of nitrate in the snow at Dome C whereas the latter formulates the physical processes involved in the co-condensation in a relatively simple and yet prognostic fashion.

The first model, which accounts only for the surface adsorption of HNO3, over-predicts the measured concentrations of nitrate in the surface snow almost all the time over the annual cycle and particularly from September to mid-November when the model over-prediction reaches a factor of eight. Uncertainties in the gaseous HNO3 concentrations in ambient air and in the model input of snow temperature and specific surface area all appear to be contradictory and/or insufficient to bring the simulated concentrations of nitrate in the snow down satisfactorily to the measured ones. The authors then elaborate on the existence of major pitfalls in the laboratory experimental data for the surface adsorption of HNO3 on ice, which I think is very useful and compelling. Hence, by using the other three models (BC1, BC2 and BC3), the authors seek possibilities for reproducing the measured nitrate concentrations by neglecting the surface adsorption. This endeavor seems to have worked out quite well. The reasonable success of all the BC1 to BC3 models during the wintertime indicates the key role of solid-state diffusion and thermodynamic equilibrium solubility of HNO3 in bulk ice during that time of the

year at Dome C. On the other hand, the failure of the BC1 model and the success of the BC2 and BC3 models during the summertime indicate the co-condensation as a major process leading to the summertime peak of nitrate concentrations in the surface snow at Dome C. The bulk incorporation models (BC1, BC2 and BC3) all have a difficulty in capturing the relatively high concentrations of surface snow nitrate from September to mid-November, which the authors interpret as an indication for the involvement of surface adsorption neglected in these models.

Overall, I find the paper very interesting and support its publication in ACP once the following problems are addressed. My biggest complaint to the current state of the manuscript is the insufficient detail of the BC3 model provided in the model description section and of its behavior in terms of water vapor condensation in the discussion section, as I mention below in the specific comments. I am also puzzled by the formulation of the BC2 model as currently described, which I wish the authors to clarity. The quality of English should be improved to catch up with the quality of science presented in the paper. To some extent, this problem may be addressed by an English copy-editor assigned by the editorial office at the final production of the manuscript. But, according to my own experience with publication process for ACP, the authors themselves should put some efforts beforehand because the copy-editor can often misinterpret loosely written sentences.

[Major comments]

1. I am puzzled by the formulation of the BC2 model as currently described. Section 3.3.2 states that equation (11) is used to diagnose the concentration of HNO3 ($X_{\text{HNO3}}$) according to its thermodynamic equilibrium solubility into ice and enhanced uptake due to co-condensation with water vapor. Does this equation apply to the concentration of HNO3 in the entire volume of the snow grain? If this is the case, the model assumes that the entire volume of the snow grain instantly feels the impact of the near-surface domain of the grain where the concentrations of HNO3 are controlled by the co-condensation, which seems inappropriate. Judging from the initial "spin-up" behavior of the BC1 model (section 5.1 and figure 4), I presume that the relevant timescale of solid-state diffusion within the bulk ice volume is on the order of a week to a month. Please clarify.

2. The authors should also provide further details of how they formulate the BC3 model and how it behaves in terms of the condensation of water vapor and its impact on the radius growth ($\Delta r$) of the snow grain. Without these pieces of information, one cannot really make sense of why this model is so successful in reproducing the summertime peak of nitrate concentrations at Dome C and are unable to discuss their potential future studies in light of the present model results. So I would like the authors to clarify the following aspects in the formulation of the model and its behavior: a) what is the magnitude of $\Delta r/\Delta t$ prescribed in the model? The authors should provide this information in a time series over the entire annual cycle in the supplement. Does $\Delta r/\Delta t$ also vary diurnally to a significant degree?; b) From the description in section 3.3.3, it seems that equation (15) is applied to prescribe the concentration of nitrate only within the outermost layer of the depth of $\Delta r$ in the snow grain. If this is the case, the model seems to be assuming that the high concentration of nitrate due to the co-condensation is locked into that layer after the model proceeds to the next time step $(t + \Delta t)$. How is the solid-state diffusion in the deeper layers of the snow grain handled in this model? This aspect of the model formulation should be properly described, perhaps with additional equations in section 3.3.3; and c) as currently formulated in the model, it appears that the evaporation of water from the snow grain does nothing to nitrate in it. But I wonder if it is appropriate to assume that such an evaporation of water may expose the layer of ice once buried with supersaturated HNO3, leading to the volatilization of HNO3 to ambient air. Is it not feasible at all to consider this possibility in the present model framework?

[Minor comments]

1. After introducing equation (8), it may be useful state the relevant timescale of solid-state diffusion with the spherical snow gain with the radius of 85 micrometers. This

information can then be restated in section 5.1, where the authors refer the initial drop of nitrate concentrations in the BC1 model from 500 ng g$^{-1}$.

2. L625-626: This statement on indication for the involvement of surface adsorption in the spring should be stressed more clearly in the abstract and conclusions.

3. L645-651: What happens to the BC1 model if you decrease the solid-state diffusivity of HNO3 by 72%?

4. Table 1: Is it useful to show mean model biases as well here?

[Editorial suggestions]

L51: features -> exhibits

L53: shows -> implies

L60-61: affinity TO

L62-63: "small chemical species" -> "small molecules"

L71: "consists in" -> is

L100: IN the late 1990's

L106: In THAT study

L167: "further accounting for" -> "a more extensive characterization of"

L171: features -> shows

L192: "Indeed, the . . ." -> "The . . ."

L196-198: Sounds awkward to me. Please consider rephrasing.

L349: adjusted -> adjustable

L352: "submitted to" -> "in accordance with"

L409: during -> over

L423: the AVERAGE modelled concentration

L425: "till end" -> "toward the end of"

L447: compelling -> forcing

L531: "the SSA value is relevant to summertime" -> "the SSA value adopted in our model is comparable to summertime observations"

L585: It is not clear to me what is meant by the statement "features similar shape".

L609: supposed TO BE constant

L625-626: "another clue to say" -> "another indication"

L681: "getting rid of" -> ignoring

L692: "1-D," -> "one-dimensional (1-D)

Supplement line 5: ARE not straightforward

Supplement line 18: WAS applied to

Supplement line 20: Drop "at total"

Supplement line 22: feature -> shows

---

## Author Comment (AC1) · 13 Aug 2016

**Reply to the review of Anonymous Referee #1**

Replies to his/her remarks and suggestions are given below. For the sake of clarity, the reviewer's comments are in blue italics and our response is in black font.

*Bock et al. use a model to examine the impacts of the physical exchange processes, adsorption, bulk diffusion and co-condensation, on the depth profiles of nitrate concentration in snow at Dome C, Antarctica. They find that bulk diffusion and co-condensation alone can explain the observed profiles.*
*This paper was not clearly written and thus very frustrating to read. I don't understand how one can model snow nitrate without including important processes such as atmospheric deposition and the photolysis of snow nitrate. The former is how nitrate gets to the snow in the first place, and the latter has been shown to be the dominant loss process of snow nitrate at Dome C.*

Anonymous Referee #1 states that the photolysis of snow nitrate is the dominant loss process of snow nitrate at Dome C, and should be included in a modelling framework dedicated to the study of snow nitrate. We are aware that some studies concluded that the photolysis is the dominant loss process. We already highlighted these results in the submitted manuscript (L92-93). However, as explained in our paper (L50-54), the dramatic increase of nitrate concentration in surface snow during the summertime implies that uptake processes have a much stronger magnitude than sinks (which is mainly the photolysis).
In order to strengthen this point, we added a calculation of photolysis and co-condensation fluxes in the Supplementary Information (Sect. 3):

In this section, an estimation of loss and uptake fluxes is presented. Both calculations are based on the following assumptions: a skin layer thickness of 3 mm, with a snow density of 0.3 kg m⁻³. The fluxes are calculated for an area of 1 cm².
The photolysis flux is calculated for a single nitrate concentration of 1200 ng g⁻¹, which results in $9.7 \times 10^{14}$ molecules in the 1 cm² × 3 mm volume. France et al. (2011) reported a photolysis rate for nitrate of about $1 \times 10^{-7}$ s⁻¹ in Dome C surface snow, for a solar zenith angle (SZA) of 52° which is the maximum solar elevation at Dome C. The resulting photolytic loss flux is $9.7 \times 10^7$ molecules cm⁻² s⁻¹.
The uptake flux resulting from the co-condensation process is calculated by assuming that the 1 cm² × 3 mm volume is filled with ice spheres of radius 85 µm (the value used in this study) up to the prescribed density. This results in ~ 37200 spheres. In the theoretical study by Dominé and Thibert (1996), the average concentration in the condensed layer immediately before another layer condensates and isolates the previous one, is given by the integral of Eq. (15):

$$X_{average} = X_{kin} + \frac{X_{eq} - X_{kin}}{\Delta r} \int_0^{\Delta r} erfc\left(\frac{x}{2\sqrt{(D \cdot t)}}\right) dx$$

Using the same input data as in the model, and assuming that this average concentration multiplied by the condensed volume corresponds to the quantity of nitrate actually taken up by the snow, we calculate an average uptake flux of $5.4 \times 10^9$ molecules cm⁻² s⁻¹ over the December 2009 to January 2010 period. The minimum and maximum values are $1.6 \times 10^8$ molecules cm² s⁻¹ and $2.7 \times 10^{10}$ molecules cm⁻² s⁻¹, respectively.
As a conclusion, the uptake flux due to the co-condensation appears to be ? 56 times larger, on average, than the photolysis flux calculated for the highest solar elevation condition. This confirms that photolysis loss can be neglected when studying the nitrate concentration in the skin layer.
We added the following paragraph in the main text to refer to this comparison:
(L665-671)
The photolysis has not been included in this study, because the dramatic increase of summer nitrate concentration in the skin layer demonstrate

that uptake processes overtake loss processes in this specific layer. In
order to refine this comparison regarding the budget of nitrate in the
skin layer, an estimation of the uptake and destruction fluxes is
presented in the supplementary information (Sect. 3). It appears that the
uptake flux calculated with the BC3 parameterisation is 1.5 orders of
magnitude larger than the maximum loss flux due to photolysis. This
confirms that photolysis loss is negligible as compared to the co-
condensation uptake when studying the skin layer concentration.

Anonymous Referee #1 states that atmospheric deposition is an important process which should be taken into
account. We are not sure of the exact processes he/she is referring to. However, we emphasise that our study
focuses on air – snow exchanges processes on the scale of a snow grain. The study of atmospheric processes
is much beyond the scope of our work. However, we also stress that  the air-snow uptake processes studied
here are able to explain "how nitrate gets to the snow in the first place".
* * *
*Although I agree that their study is important, as such physical processes will influence the distribution of
nitrate in the snow column and the snow grain, the latter of which may influence e.g., how photolabile snow
nitrate is, I don't see how they can ignore these other important processes. It seems that it would be better to
use their model to examine the results in a laboratory, where the processes they ignore can be controlled.*
*Since the manuscript is not clearly written, it is possible that I am misunderstanding something important
about their modeling framework*

We agree with Anonymous Referee #1 that the developed modelling framework allows us to infer the
distribution of nitrate inside the snow grain, which is likely to influence how photolabile snow nitrate is. We
emphasised this aspect in the conclusion by adding the following sentence:
(L743-749)
In this study focused on skin layer snow, nitrate photolysis inside the
snow grain has not been implemented since nitrate loss is much weaker
than uptake for this specific layer, as demonstrated by the dramatic
increase of nitrate concentration during summer. This is not true for the
whole snowpack, and photolysis should be included in a 1-D snow chemistry
model. For that purpose, the description of a snow grain as a layered
medium will enable using different quantum yields, after some studies
suggested that it span more than 2 orders of magnitude depending on the
availability of nitrate inside the ice matrix (Zhu et al. 2010, Meusinger
et al, 2014).

Cited Papers:
Dominé, F. and Thibert, E.: Mechanism of incorporation of trace gases in ice grown from the gas phase, Geophysical
Research Letters, 23, 3627–3630, doi:10.1029/96GL03290, 1996.

France, J. L., King, M. D., Frey, M. M., Erbland, J., Picard, G., Preunkert, S., MacArthur, A., and Savarino, J.: Snow
optical properties at Dome C (Concordia), Antarctica; implications for snow emissions and snow chemistry of reactive
nitrogen, Atmospheric Chemistry and Physics, 11, 9787–9801, doi:10.5194/acp-11-9787-2011, 2011.

Meusinger, C., Berhanu, T. A., Erbland, J., Savarino, J., and Johnson, M. S.: Laboratory study of nitrate photolysis in
Antarctic snow. I. Observed quantum yield, domain of photolysis, and secondary chemistry, The Journal of Chemical
Physics, 140, 244 305, doi:10.1063/1.4882898, 2014.

Zhu, C., Xiang, B., Chu, L. T., and Zhu, L.: 308 nm photolysis of nitric acid in the gas phase, on aluminum surfaces,
and on ice films, The Journal of Physical Chemistry A, 114, 2561–2568, doi:10.1021/jp909867a, 2010.

---

## Author Comment (AC2) · 13 Aug 2016

**Reply to the review of Anonymous Referee #2**

We thank Reviewer #2 for his/her positive appreciation of this work and especially his/her detailed comments regarding the model description and the discussion of the co-condensation process and parameterisation.
Replies to his/her remarks and suggestions are given below. For the sake of clarity, the reviewer's comments are in blue italics and our response is in black font.

*Overall, I find the paper very interesting and support its publication in ACP once the following problems are addressed. My biggest complaint to the current state of the manuscript is the insufficient detail of the BC3 model provided in the model description section and of its behavior in terms of water vapor condensation in the discussion section, as I mention below in the specific comments. I am also puzzled by the formulation of the BC2 model as currently described, which I wish the authors to clarity. The quality of English should be improved to catch up with the quality of science presented in the paper. To some extent, this problem may be addressed by an English copy-editor assigned by the editorial office at the final production of the manuscript. But, according to my own experience with publication process for ACP, the authors themselves should put some efforts beforehand because the copy-editor can often misinterpret loosely written sentences.*

We asked native English speakers to proofread the manuscript. We think that the quality of English now match the requirements for a publication in ACP.

*[Major comments]*
*1. I am puzzled by the formulation of the BC2 model as currently described. Section 3.3.2 states that equation (11) is used to diagnose the concentration of HNO₃ ($X_{HNO_3}$) according to its thermodynamic equilibrium solubility into ice and enhanced uptake due to co-condensation with water vapor. Does this equation apply to the concentration of HNO₃ in the entire volume of the snow grain? If this is the case, the model assumes that the entire volume of the snow grain instantly feels the impact of the near-surface domain of the grain where the concentrations of HNO₃ are controlled by the co-condensation, which seems inappropriate. Judging from the initial "spin-up" behavior of the BC1 model (section 5.1 and figure 4), I presume that the relevant timescale of solid-state diffusion within the bulk ice volume is on the order of a week to a month. Please clarify.*

The second configuration of the model describes the snow grain as a layered sphere. Solid state diffusion is the only exchange process occurring between adjacent layers. Lying at the interface, the outermost layer undergoes both air – snow exchange processes and solid state diffusion with the inner layer. We developed three parameterisations to describe the air – snow partitioning and prescribe the nitrate concentration in the outermost layer. It is the boundary condition (BC) of the diffusion scheme, and this is why we choose to label the different parameterisations BC1 to BC3. This was stated in the introduction of section 3.3 (L289-294).
The entire volume of the snow grain is thus never constrained, it only "feels the impact" of the outermost layer concentration through solid state diffusion.

To clarify the description of these parameterisations, we changed the text as follows:
- we split section 3.3.1 ("Diffusion scheme and equilibrium boundary condition (BC1)") into 2 sections ("3.3.1 Diffusion scheme"; "3.3.2 Equilibrium boundary condition (BC1)"), and we stressed that the diffusion scheme applies to any of the chosen boundary conditions.
- we slightly rephrased the description of BC2 to emphasise that the developed parameterisation only changes the outermost layer concentration, as compared to BC1:

```
[old text] Thus, in order to test these hypotheses, a first simple
diagnostic parameterisation of cocondensation
process was implemented by adding an adjustable term in the boundary
condition definition of the concentration:
```

→

[new text] Thus, in order to test these hypotheses, a first simple diagnostic parameterisation of co-condensation process was implemented by adding an adjustable term to prescribe the outermost layer concentration (BC2):
(…)
Solid state diffusion within the layered snow grain then proceeds as previously described (Sect. 3.3.1).

*2. The authors should also provide further details of how they formulate the BC3 model and how it behaves in terms of the condensation of water vapor and its impact on the radius growth (Δr) of the snow grain. Without these pieces of information, one cannot really make sense of why this model is so successful in reproducing the summertime peak of nitrate concentrations at Dome C and are unable to discuss their potential future studies in light of the present model results. So I would like the authors to clarify the following aspects in the formulation of the model and its behavior:*
*a) what is the magnitude of Δr/Δt prescribed in the model? The authors should provide this information in a time series over the entire annual cycle in the supplement. Does Δr/Δt also vary diurnally to a significant degree?;*

Following the request of Anonymous Reviewer #2, we added a new figure (reproduced below) showing the time series of the radius growth rate. We decided to include it in the main text since it clearly shows that this time series feature similar yearly pattern that the skin layer nitrate concentration. This is another evidence that the skin layer concentration is driven by temperature gradient metamorphism, and co-condensation. We also added the following paragraphs in the discussion about BC3 results (Sect. 5.3).
(L630-642)
The radius growth rate $\Delta r/\Delta t$ as derived from Eq. (14) is presented in Fig. 5. It spans roughly three orders of magnitude over the year, from about $10^{-12}$ m s$^{-1}$ in winter to ~8 $10^{-10}$ m s$^{-1}$ in summer. The explanation of this behaviour is twofold. First, the diurnal temperature cycle has a larger amplitude in summer, which enhances the temperature gradient close to the surface. Second, the vapour pressure over ice increases exponentially with temperature. As a consequence, with a given temperature gradient, the gradient of water vapour concentration used in Eq. (12) is larger if temperatures are higher. This also explains the diurnal variation of the grain radius growth. The most striking feature of the radius growth rate is that it peaks during the same period of the year that the peak of nitrate concentration in the skin layer. The yearly pattern of the radius growth rate predicted by our model is also consistent with independent studies focused on snow physical properties (Picard et al, 2012, Libois et al 2015). This comes as another evidence that snow metamorphism, and co-condensation, have a major influence over the snow chemical concentration.

[Figure]

(L645-650)
A diurnal variation of the modelled concentration is observed, as a consequence of the diurnal variation of the radius growth rate. However, the diurnal variation of the concentration is much smoother because solid state diffusion in the whole snow grain softens the large diurnal

variations in the outermost layer of the snow grain. The modelled diurnal
variation of the concentration is smaller than 20 %, which is similar to
the measurements uncertainty due to spatial heterogeneity.

*b) From the description in section 3.3.3, it seems that equation (15) is applied to prescribe the concentration of nitrate only within the outermost layer of the depth of Δr in the snow grain. If this is the case, the model seems to be assuming that the high concentration of nitrate due to the co-condensation is locked into that layer after the model proceeds to the next time step (t+Δt ). How is the solid-state diffusion in the deeper layers of the snow grain handled in this model? This aspect of the model formulation should be properly described, perhaps with additional equations in section 3.3.3;*

In model configuration 2 (handling solid state diffusion), the outermost layer concentration is first prescribed according to one of the three different boundary conditions (BCs) developed in this study. Then solid state diffusion proceeds as described in the relevant section (Sect. 3.3.1). Whatever the BC, the concentration is thus never "locked into" the outermost layer.
We expect that the improved description of the model configuration 2 (see our answer to Major comment 1) also addresses this comment regarding the clarity of the description for BC3.
We also added a sentence at the end of the section describing BC3 to clarify the role of the radius increase calculation:
(L429-432)
We emphasise that the radius of the modelled snow grain is kept unchanged
along the whole simulation. The calculation of the radius increase due to
the condensation of water vapour is only used to compute the
concentration (Eq. 15) at the surface of the modelled snow grain (BC).

*c) as currently formulated in the model, it appears that the evaporation of water from the snow grain does nothing to nitrate in it. But I wonder if it is appropriate to assume that such an evaporation of water may expose the layer of ice once buried with supersaturated HNO3, leading to the volatilization of HNO3 to ambient air. Is it not feasible at all to consider this possibility in the present model framework?*

We agree with Reviewer #2 that sublimation of water is an important process regarding the mass balance of a snow grain. Despite many efforts to integrate this process in our model, we have not found a satisfactory way to implement it. However, we stress that in the current hypothesis, the snow grain has a constant radius. The mass growth calculated with Eq. (12) in order to estimate the thickness $\Delta r$ of a virtually condensed layer (Eq. 14) is only used to calculate the theoretical concentration in the condensed layer according to Eq. (15). We have stated this feature more clearly, as presented previously (answer to major comment 2b).
A model able to describe both the condensation and the sublimation, with a varying radius of the snow grain, would obviously be a better way to proceed. To the best of our knowledge, only 2-D or 3-D studies, using either modelling or microtomography, are currently able to provide a satisfactory description of these processes inside the snowpack. This requires knowing the temperature field within both the snow grains and the interstitial air, in order to calculate both condensation and sublimation (see for instance Fig. 5 in Calonne et al 2015, or the studies by Calonne et al 2014 and Ebner et al 2016). Such accurate modelling is even more difficult when considering the skin layer, since the processes governing the water vapour exchange with the lower troposphere are different than those occurring only in the porous snowpack. We are not aware of any study extending the domain to the interface with the atmosphere.

We also agree with Anonymous Reviewer #2 that not taking into account the sublimation might lead to important errors if supersaturated layers buried inside the ice matrix become exposed to the interstitial air and sublimate. This would indeed lead to an important loss regarding the overall snow concentration. However, this does not happen in our model, since the radius is kept unchanged. In other words, the supersaturated condensed layer is virtually condensed at the beginning of each timestep, and virtually removed at the end of each timestep. During the timestep, the surface of the snow grain sees the concentration as calculated with Eq. (15), and exchanges with the core of the snow grain through solid state diffusion.

The comparison of BC1 and BC3 (Fig. 4 and 5) also reveals that winter concentration is mostly unchanged between these two parameterisations. In wintertime, the radius increase calculated with Eq. (14) is so small compared to the typical diffusion length that the argument of the erfc function in Eq. (15) tends towards 0, thus the prescribed concentration tends towards $X_{eq}$. We believe that the consistent behaviour of the model throughout the year of simulation also demonstrates that there is no undue accumulation when using BC3, which would necessary lead to a long term drift if this was not the case.

*[Minor comments]*
*1. After introducing equation (8), it may be useful state the relevant timescale of solid-state diffusion with the spherical snow gain with the radius of 85 micrometers. This information can then be restated in section 5.1, where the authors refer the initial drop of nitrate concentrations in the BC1 model from 500 ng g$^{-1}$.*
We added the following statement after Eq. 8 (numbered 7 in the revised manuscript):
(L302-309)

```
The modelled snow surface temperature ranges from 198 K to 253 K (average
222 K) during the studied period. The diffusion coefficient thus ranges
from  8.9×10⁻¹⁸  m² s⁻¹  to  6.4×10⁻¹⁵  m² s⁻¹  (average  7.1×10⁻¹⁶  m² s⁻¹).  A
characteristic  time  for  diffusion,  τ,  can  be  estimated  as  τ = l² / D
where  l  is  a  characteristic  diffusion  length.  Considering  the  assumed
spherical  geometry  of  the  snow  grain,  when  diffusion  reaches   0.21×R ,
50 %  of  the  volume  is  affected;  and  when  diffusion  reaches   0.37×R ,
75 %  of  the  volume  is  affected.  Using  these  values  as  characteristic
diffusion    length    and    the    average    diffusion    coefficient,    the
characteristic  times  for  diffusion  are   τ.₅₀≃5   days  and   τ.₇₅≃16  days.
```

We rephrased the end of the sentence in Sect. 5.1 to refer to this characteristic diffusion time:
(L578-580)

```
The time needed to re-equilibrate the snow grain concentration, roughly 2
weeks, compares well with the characteristic diffusion time (see Sect.
3.3.1).
```

*2. L625-626: This statement on indication for the involvement of surface adsorption in the spring should be stressed more clearly in the abstract and conclusions.*
The analysis provided in conclusion of Sect. 4.2 (L551-565 of the original manuscript) is intended to give a rough estimation of the error when ignoring the adsorption process, given the inability of the current parameterisation to fit the measurements. However, we stressed that assuming a constant overestimation factor is a strong hypothesis, and the conclusion of this analysis should be used with caution. We thus only mentioned it in the conclusion in the revised manuscript:
(L725-729)

```
Assuming that the adsorption parameterisation is flawed by a constant
overestimation factor which would leave the yearly pattern unchanged, the
maximum featured by the modelled adsorbed concentration in September and
October suggests that adsorbed nitrate might account for roughly 30 % of
snow nitrate during these 2 months. As for the rest of the year and based
on the same hypothesis, adsorbed nitrate should account for less than
10 % of snow nitrate.
```

*3. L645-651: What happens to the BC1 model if you decrease the solid-state diffusivity of HNO3 by 72%?*

When reducing the diffusion coefficient by 72 % in BC1 configuration, the modelled concentration features smoother variations. The RMSE is also slightly reduced from 437 ng g$^{-1}$ to 431 ng g$^{-1}$.

[Figure]

*4. Table 1: Is it useful to show mean model biases as well here?*

The numerical values of the RMSE for the various runs are only shown in Table 1, and not mentioned in the main text. We believe that it can help the reader to appreciate the improvements in the reproduction of the measurements along with the various runs. We also think that it could be useful to compare with forthcoming studies based on similar datasets, and/or using similar modelling framework.

However, if the Anonymous Reviewer #2 and/or the Editor want this information to be removed from Table 1, we agree it is not essential.

*[Editorial suggestions]*

All editorial suggestions have been modified accordingly.

Cited Papers:

Calonne, N., Flin, F., Geindreau, C., Lesaffre, B. & Rolland du Roscoat, S. Study of a temperature gradient metamorphism of snow from 3-D images: time evolution of microstructures, physical properties and their associated anisotropy. *The Cryosphere* **8,** 2255–2274 (2014).

Calonne, N., Geindreau, C. & Flin, F. Macroscopic modeling of heat and water vapor transfer with phase change in dry snow based on an upscaling method: Influence of air convection. *Journal of Geophysical Research: Earth Surface* **120,** 2476–2497 (2015).

Dominé, F. & Thibert, E. Mechanism of incorporation of trace gases in ice grown from the gas phase. *Geophys. Res. Lett.* **23,** 3627–3630 (1996).

Ebner, P. P., Schneebeli, M. & Steinfeld, A. Metamorphism during temperature gradient with undersaturated advective airflow in a snow sample. *The Cryosphere* **10,** 791–797 (2016).

Gallet, J.-C., Domine, F., Savarino, J., Dumont, M. & Brun, E. The growth of sublimation crystals and surface hoar on the Antarctic plateau. *The Cryosphere* **8,** 1205–1215 (2014).

---

## Author Response (AR2)

Dear Professor V. Faye McNeill,

We have now fully addressed all the comments you and both anonymous reviewers made. We agree with the latest remarks and advice.

The version submitted on the 13th of August already included all the changes to the manuscript that took into account the comments and suggestions of the anonymous reviewers.

We now submit a further revised version which takes into account your comments and suggestions. The major changes of this last version are the following:

- a reorganisation of the introduction, now split into 3 subsections, in order to clarify and to answer the question about aerosol processes;
- the comparison of loss and uptake fluxes now features in the main text (Sect. 5.3) rather than in the supplementary information, and also includes a comparison with available measurements.

We also improved the quality of English of the manuscript.

At the end of this document, we included two track-change versions:

- a comparison with the revised version submitted in August, in order to highlight the recent changes to answer your comments and suggestions (pp. 3 - 41);

- a comparison with the original version to show all changes done following your and the anonymous reviewers' remarks and suggestions (pp. 42 - 80).

A detailed reply and argumentation to your remarks and suggestions are given below. For clarity, we kept your comments in blue and italic while our responses are in black font. We hope that you will be satisfied with our latest improvements and that the manuscript now agrees with the high quality required for publication in ACP. We sincerely thank the editorial office for the comments which allowed us to significantly improve our manuscript. The co-editor and reviewers are thanked in the acknowledgement section for their very constructive work.

With our best regards

**Reply to the editor**

Comments to the Author:

I believe that this manuscript represents a valuable contribution to the field and will eventually be publishable. However, I agree with Anonymous Referee #1 in that a full discussion, in the main manuscript, of other possible sources and sinks of nitrate which are not accounted for in the models, and how they compare to the sources and sinks currently represented in the models (i.e., why you believe they are insignificant for this system), is warranted. I believe that by "deposition" the referee is referring to aerosol deposition, since you of course are considering different mechanisms of gas deposition here.

We agree with Prof. McNeill and Anonymous Referee #1 that aerosols dry deposition is not accounted for, for two reasons:

- on the one hand, the study of atmospheric processes relevant to the atmospheric column scale are clearly beyond the scope of our study;

- on the other hand, as described in Sect. 2.1.1 (Atmospheric nitrate), particulate nitrate (i.e. nitratecontaining aerosols), if present, represents a minor fraction (10 - 30 %) of atmospheric nitrate. Thus, it is assumed that the measured atmospheric nitrate is in the gas phase.

In order to clarify the various sources and sinks of nitrate in the studied skin layer, we reorganised the introduction with a specific section to detail the relevant processes, and their significance. We also clarified that aerosol-related processes are not taken into account in our study, because of the very low amount of aerosols in Dome C.

The discussion of nitrate photolysis should be expanded upon and placed in the context of much more sophisticated treatments of nitrate photolysis in Antarctic snow in the literature (e.g. Boxe and Saiz-Lopez, ACP 2008 and Shi et al. Atmos. Chem. Phys., 15, 9435–9453, 2015 www.atmos-chem-phys.net/15/9435/2015/). And this discussion belongs in the main manuscript, not the supplement. We agree that the comparison between uptake (co-condensation) and loss (photolysis) fluxes

belongs to the main manuscript, and we moved the discussion to the appropriate Section (Sect. 5.3). Our model focuses solely on the skin layer, and in that respect it is a 0-dimension model (or a box

model), with a unique snow grain surrounded with interstitial air. The snow grain is described as a layered sphere, and solid state diffusion is computed following 1-dimensional diffusion equation. In order to remove a potential source of confusion, we removed the comparison of summer vs. winter nitrate concentration profiles in Section 2.1.2 (Snow nitrate), and the accompanying Figure 2, which had only an illustrative purpose, to broaden the description of the nitrogen recycling within the snowpack.

Since our study focuses on the skin layer, and the photolysis flux calculation is done locally, it is not possible to compare this flux with measurements or 1-D models, which integrate the photolysis flux over the whole photic zone of the snowpack. However, the calculation of the photolysis flux emanating from the sole skin layer is nothing more than a multiplication of the the nitrate photolysis rates provided by France et al (2011) times the available nitrate in a control volume. Thus, we believe that a comparison with other data is not mandatory.

On the other hand, the calculation of the uptake flux ascribed to the co-condensation process can be compared to atmospheric fluxes of  $HNO_3$  towards the snow, which corresponds to our modelling framework. Such a comparison is relevant, since the uptake flux calculation involved the whole developed parameterisation. We added a comparison of the computed uptake flux with measurements found in the literature.

Track change version 1:

Differences between the latest version and the version submitted on the 13th of August.

This track-change version highlights the improvements following the co-editor comments and suggestions.

[revised manuscript text omitted]
{3}{\frac{SSA \times \rho_{\text{ice}}}{SSA \rho_{\text{ice}}}} \frac{3}{\frac{SSA \rho_{\text{ice}}}{SSA \rho_{\text{ice}}}}$$
(1)

where R is the radius (in m), SSA is the snow specific surface area (in m2 kg-1) and  $\rho_{ice}$  is the ice density, with  $\rho_{ice} \simeq 924 \text{ kg m}^{-3}$  (Hobbs, 1974, at -50 °C, DC annual mean temperature). When this study was initiated, the only SSA value reported at DC was 38.1 m2 kg-1 for the first centimetre, decreasing monotonically to 13.6 m2 kg-1 at 70 cm depth (Gallet et al., 2011, Fig. 4 and Table A1).

- Recent work specifically studying surface hoar at DC reported very close values, with an average of 39.0 m2 kg-1 for the top first centimetre of snow, and 26.4 m2 kg-1 for the second centimetre (Gallet et al., 2014). Thus, SSA was set to a value of 38.1 m2 kg-1 by default in the model, leading to a grain radius *R* = 85 µm. Recently Libois et al. (2015) and Picard et al. (2016) investigated seasonal variations of SSA at DC showing that these values are typical of the summer while 2 to 3-fold higher
  values are observed in winter. The effect of changing SSA was further tested in a sensitivity test
- values are observed in winter. The effect of changing SSA was further tested in a sensitivity test presented in Sect. 5.4.

**3 Model description**

**3.1 From gaseous HNO3 to solid solution of nitrate in snow**

A brief summary of the current knowledge about solvation steps which lead gaseous HNO3 to form solid solution in bulk ice is presented in this section.

The uptake of trace gases on ice, and more specifically of acidic gases among which  $HNO_3$ , has been the subject of considerable investigation numerous investigations (see reviews by Abbatt, 2003; Huthwelker et al., 2006). Conceptually, this uptake proceeds firstly by molecular adsorption of  $HNO_3$ , followed by the ionisation (or dissociation) and then progressive solvation at the surface

295

5 leading to a partial solvation shell (Buch et al., 2002; Bianco et al., 2007, 2008). In a second stage, thought to be much slower, the adsorbed nitrate anions sink into the innermost crystal layers, leading to a complete solvation shell, and diffuse towards the bulk crystal. Recent studies addressed the ionisation state of HNO3 adsorbed on ice surface, either using surface sensitive spectroscopy techniques

(Křepelová et al., 2010; Marchand et al., 2012; Marcotte et al., 2013, 2015) or through molecular

- 300 dynamics models (Riikonen et al., 2013, 2014). Molecular adsorbed state is found to be metastable, which happens only at very low temperatures (45 K), whilst ionic dissociation occurs irreversibly irreversibly occurs upon heating at 120 K (Marchand et al., 2012). Molecular dynamics simulations suggest a pico and subpicosecond ionisation of HNO3 in the defects sites (Riikonen et al., 2013), further supporting that molecular adsorption of HNO3 on ice is a fleeting state prior to ionisation, at
- 305 least for environmentaly relevant temperatures.

Despite these recent improvements in the understanding of  $HNO_3$  ionisation following adsorption on an ice surface, the transition between surface (adsorption) and bulk (diffusion) processes still needs to be fully characterised. To the best of our knowledge, no process-scale parameterisation of the dissociation/solvation exists at the moment. Such parameterisation would be necessary to

310 link surface and bulk concentrations, and further studies are thus needed to fully characterise the transition between these states. For this reason, both processes were treated separately in our model. The model configuration 1 (adsorption) is described in the next section, while the configuration 2 (solid state diffusion) is described in Sect. 3.3.

**3.2 Model configuration 1: adsorption**

315 The HNO3 surface coverage is a function of temperature and pressure only. Crowley et al. (2010) presented a compilation of data evaluated by a IUPAC subcommittee, that characterises heterogeneous processes on the surface of solid particles, including ice. They recommend the use of a single-site Langmuir isotherm which gives the fractional surface coverage  $\theta$ :

$$\theta = \frac{N}{N_{\text{max}}} = \frac{K_{\text{LangP}} P_{\text{HNO}_3}}{1 + K_{\text{LangP}} P_{\text{HNO}_3}}$$
(2)

320 where  $N_{\text{max}} = 2.7 \times 10^{18}$  molecules m-2 is the HNO3 surface coverage at saturation,

$$K_{\text{LangP}} = \frac{K_{\text{LinC}} \mathcal{N}_{\text{A}}}{N_{\text{max}} \underline{R} \underline{\mathcal{R}} T} \text{ (in Pa}^{-1})$$
(3)

$$K_{\rm LinC} = 7.5 \times 10^{-7} \exp\left(\frac{4585}{T}\right) ({\rm in \ m})$$
 (4)

KLangP and KLinC are partition coefficients expressed in different units, N is the HNO3 surface coverage (in molecules m-2), PHNO3 is the HNO3 partial pressure (in Pa), NA is the Avogadro constant,
T the snow temperature (in K) and R-R the molar gas constant (R = 8.314 R = 8.314 J K-1 mol-1).

This parameterisation is established for temperatures ranging from 214 K to 240 K, and is used here at which is almost adequate to DC temperatures, typically in the 200–250 K range - (see Fig. 1b). The conversion of surface coverage to bulk concentration is done using the SSAvalueSSA:

$$[HNO_3] = \frac{N \times SSA}{\mathcal{N}_A} \tag{5}$$

330 where  $[HNO_3]$  is the nitrate concentration (in mol m-3).

The results and discussion following adsorption calculation are presented in Sect. 4.

**3.3 Model configuration 2: solid state diffusion**

In configuration 2, the model computes solid state diffusion in a layered snow grain. The outermost layer concentration or boundary condition (BC) was-is successively set according to three distinct

335 parameterisations. Firstly, the NO3- concentration at the air – ice interface was is set according to thermodynamic equilibrium (BC1). In a second stage, the kinetic, co-condensation process was is taken into account through an empirical, diagnostic parameterisation (BC2), then with . Then, using the results from the previous BCs, a physically based prognostic parameterisation is developed (BC3). The general diffusion scheme and specific BCs are presented in the next sections.

**340 3.3.1 Diffusion scheme**

In configuration 2, the model considers a spherical snow grain with a radius  $R = 85 \,\mu\text{m}$ , divided in concentric layers of constant thickness  $\delta r = 0.05 \,\delta R = 0.05 \,\mu\text{m}$ . The model computes the solid state diffusion equation in spherical geometry with radial symmetry in the snow grain:

$$\frac{\partial C(r,t)}{\partial t} = D\left(\frac{2}{r}\frac{\partial C(r,t)}{\partial r} + \frac{\partial^2 C(r,t)}{\partial r^2}\right) \tag{6}$$

345 where C(r,t) is nitrate concentration in the layer of radius r at time t, and D is the diffusion coefficient of HNO3 in ice provided by Thibert and Dominé (1998):

$$D = 1.37 \times 10^{-4} \times 10^{-2610/T} \text{ (in m}^2 \text{ s}^{-1)}$$
(7)

The modelled snow surface temperature ranges from 198 K to 253 K (average 222 K) during the studied period. The diffusion coefficient thus ranges from 8.9×10-18 m2 s-1 to 6.4×10-15 m2 s-1
(average 7.1×10-16 m2 s-1). A characteristic time for diffusion, τ, can be estimated as τ = l2/D where l is a characteristic diffusion length. Considering the assumed spherical geometry of the snow grain, when diffusion reaches 0.21×R, 50% of the volume is affected; and when diffusion length and the average diffusion coefficient, the characteristic times for diffusion are τ.50 ≈ 5 days and τ.75 ≈ 16 days.

Thibert and Dominé (1998) indicated an uncertainty of  $\pm 60$  % for the diffusion coefficient, further explaining that it is probably the upper limit because of the existence of diffusion short pathways. The study by Thibert and Dominé (1998) was carried out at temperatures ranging from -8 °C to -35 °C. Nevertheless, Eq. (7) is applied to the temperature temperatures of DC surface snow, potentially leading to an increased additional uncertainty.

360

The concentration of the outermost layer of the modelled snow grain, which is the boundary condition (BC) of the diffusion equation (6), was successively parameterised in 3 different ways that are detailled in the next sections.

**3.3.2 Equilibrium boundary condition (BC1)**

365 In a first attempt labelled BC1, the outermost layer concentration was set according to the thermodynamic equilibrium solubility of HNO3 in solid solution as measured by Thibert and Dominé (1998):

$$X_{\rm HNO_3}^0 = 2.37 \times 10^{-12} \exp\left(\frac{3532.2}{T}\right) P_{\rm HNO_3}^{1/2.3}$$
(8)

where  $X_{\text{HNO}_3}^0$  is the molar fraction of HNO3 in ice, T is the snow temperature (in K) and  $P_{\text{HNO}_3}$  is the HNO3 partial pressure (in Pa).

Thibert and Dominé (1998) indicated an uncertainty of  $\pm 20$  % for equilibrium solubility. The study by Thibert and Dominé (1998) was carried out at temperatures ranging from -8 °C to -35 °C. Nevertheless, as with the diffusion coefficient, Eq. (8) is also applied to the temperature of DC surface snow temperatures, potentially leading to an increased additional uncertainty.

The results and discussion of the modelling of nitrate concentration in surface snow using this BC1 approach are presented in Sect. 5.1. We also investigated how the uncertainties over the solubility and the diffusion coefficient affect the simulations, in a sensitivity study presented in Sect. 5.4.

**3.3.3 Diagnostic co-condensation parameterisation (BC2)**

To investigate the concentration of the growing phase, an empirical, diagnostic parameterisation of the co-condensation process was firstly developed. The main purpose of this diagnostic parameterisation is to investigate the composition of the growing phase.

Valdez et al. (1989) carried out experiments on  $SO_2$  incorporation into ice growing from water vapour, and reported that the amount of sulfur incorporated into the ice increased linearly with the amount of ice deposited. Jacob and Klockow (1993) compared the concentration of  $H_2O_2$  in

- 385 the gas phase and in the snow during fog events, and showed that the molar fraction of hydrogen peroxide,  $X_{\text{H}_2\text{O}_2}$ , resulting from co-condensation was similar to the ratio of partial pressures:  $X_{\text{H}_2\text{O}_2} \simeq \frac{P_{\text{H}_2\text{O}_2}}{P_{\text{H}_2\text{O}}}$ , as previously hypothesised by Sigg and Neftel (1988). Dominé et al. (1995) refined this analysis using the kinetics theory of gases to include the number of collisions, and further taking into account the surface accommodation coefficients  $\alpha$ . They proposed that the molar fraction
- 390 of a gas  $i(X_i)$  condensating along with water vapour should obey the following equation, where M is the molar mass:

370

$$X_i = \frac{P_i}{P_{\rm H_2O}} \frac{\alpha_i}{\alpha_{\rm H_2O}} \sqrt{\frac{M_{\rm H_2O}}{M_i}}$$
(9)

However, Ullerstam and Abbatt (2005) carried out laboratory measurements of HNO3 concentration in growing ice, and their results suggested that HNO3 concentration was proportional to P0.56HNO3
 and independent of the water vapour partial pressure:

$$\log_{10}(X_{\rm HNO_3}) = 0.56 \times \log_{10}(P_{\rm HNO_3}) - 3.2 \tag{10}$$

where the exponent factor 0.56 could be explained by acid dissociation during co-condensation. Another possible explanation proposed by Ullerstam and Abbatt (2005) is that thermodynamic solubility governs at least partially the composition of a growing crystal as HNO3 is sufficiently volatile and mobile to be excluded from the growing ice. Indeed, the power 0.56 dependence to HNO3 partial

400

To summarise the conclusions of these studies, the co-condensed phase has a concentration which depends on (i) the studied trace gas partial pressure (but without agreement on the exponent in the case of  $HNO_3$ ) and (ii) may or may not depend on the water vapour partial pressure. Thus, in order

to test these hypotheses, a first simple diagnostic parameterisation of co-condensation process was

pressure is close to that of thermodynamic equilibrium solubility (in Eq. (8),  $1/2.3 \simeq 0.43$ ).

405

implemented by adding an adjustable term to prescribe the outermost layer concentration (BC2):

$$X_{\rm HNO_3} = X_{\rm HNO_3}^0 + \alpha \cdot P_{\rm HNO_3}^\beta \cdot P_{\rm H_2O}^\gamma$$
(11)

[revised manuscript text omitted]